# Histology and Immunohistochemistry of Adipose Tissue: A Scoping Review on Staining Methods and Their Informative Value

**DOI:** 10.3390/cells14120898

**Published:** 2025-06-14

**Authors:** Tom Schimanski, Rafael Loucas, Marios Loucas, Oliver Felthaus, Vanessa Brébant, Silvan Klein, Alexandra Anker, Konstantin Frank, Andreas Siegmund, Andrea Pagani, Sebastian Geis, Sophia Theresa Diesch, Andreas Eigenberger, Lukas Prantl

**Affiliations:** 1Department of Plastic, Hand and Reconstructive Surgery, University Hospital Regensburg, 93053 Regensburg, Germany; tom.schimanski@stud.uni-regensburg.de (T.S.); oliver.felthaus@ukr.de (O.F.); vanessa.brebant@ukr.de (V.B.); alexandra.anker@ukr.de (A.A.); konstantin.frank@ukr.de (K.F.); andreas.siegmund@klinik.uni-regensburg.de (A.S.); andrea.pagani@ukr.de (A.P.); sebastian.geis@ukr.de (S.G.); sophia.diesch@ukr.de (S.T.D.); andreas.eigenberger@ukr.de (A.E.); lukas.prantl@ukr.de (L.P.); 2Clinic of Plastic, Aesthetic, and Reconstructive Surgery, Döbling Private Hospital, 1090 Vienna, Austria; marios.loucas@hotmail.com

**Keywords:** histology, immunohistochemistry, fat grafting, stromal vascular fraction, extracellular matrix, regenerative medicine, staining protocol, lipoaspirate

## Abstract

Background: Histological and immunohistochemical analyses of adipose tissue are essential for evaluating the quality and functionality of lipoaspirates in regenerative medicine and fat grafting procedures. These methods provide insights into tissue viability, cellular subtypes, and extracellular matrix (ECM) composition—all factors influencing graft retention and clinical outcomes. Purpose: This scoping review aims to summarize the most commonly used staining methods and their applications in the histology and immunohistochemistry of adipose tissue. By exploring qualitative and quantitative markers, we seek to guide researchers in selecting the appropriate methodologies for addressing experimental and translational research. Methods: A systematic search was conducted using PubMed, Ovid, and the Cochrane Library databases from inception to 2024, employing Boolean operators (“lipoaspirate” OR “fat graft” OR “gauze rolling” OR “decantation” OR “coleman fat” OR “celt” OR “nanofat” OR “lipofilling” OR “human fat” AND “histol*”). Studies were included if they utilized histology or immunohistochemistry on undigested human adipose tissue or its derivatives. The inclusion criteria focused on peer-reviewed, English-language studies reporting quantitative and qualitative data on adipose tissue markers. Results: Out of 166 studies analyzed, hematoxylin–eosin (H&E) was the most frequently employed histological stain (152 studies), followed by Masson Trichrome and Sudan III. Immunohistochemical markers such as CD31, CD34, and perilipin were extensively used to distinguish stromal vascular fraction (SVF) cells, adipocytes, and inflammatory processes. Studies employing semiquantitative scoring demonstrated enhanced comparability, particularly for fibrosis, necrosis, and oil cyst evaluation. Quantitative analyses focused on SVF cell density, mature adipocyte integrity, and ECM composition. Methodological inconsistencies, particularly in preparation protocols, were observed in 25 studies. Conclusions: This review highlights the critical role of histological and immunohistochemical methods in adipose tissue research. H&E staining remains the cornerstone for general tissue evaluation in the clinical context, while specialized stains and immunohistochemical markers allow for detailed analyses of specific cellular and ECM components in experimental research. Standardizing preparation and evaluation protocols will enhance interstudy comparability and support advancements in adipose tissue-based therapies.

## 1. Introduction

Adipose tissue is a critical component in regenerative medicine, particularly in esthetic and reconstructive procedures such as lipofilling and fat grafting. These techniques leverage the unique regenerative potential of adipose-derived stem cells (ADSCs) to achieve functional and esthetic restoration [1,2]. As minimally invasive procedures, they have garnered significant attention in recent years due to their capacity to enhance tissue volume, improve skin quality, and promote long-term graft retention [3,4]. Despite these advantages, clinical outcomes often remain unpredictable, with complications such as fibrosis, necrosis, oil cyst formation, and inflammation frequently compromising success [5,6,7]. A detailed understanding of the structural and cellular properties of adipose tissue is essential for optimizing graft quality and procedural outcomes. Histological analysis has proven indispensable in this context, enabling researchers to evaluate both qualitative markers (e.g., fibrosis, necrosis, inflammation) [8,9] and quantitative markers (e.g., SVF cell counts, mature adipocyte viability, ECM composition) [10,11] that reflect tissue integrity and regenerative potential. By combining histological and immunohistochemical techniques, researchers can comprehensively assess the tissue’s functional capacity and its response to processing and transplantation [12].

Immunohistochemical staining techniques further enhance the scope of analysis by enabling the precise identification and localization of specific cell subpopulations within adipose tissue [13,14]. For example, endothelial progenitor cells (EPCs), pericytes, and ADSCs—key components of the SVF—can be identified and quantified to evaluate their role in neovasculogenesis, graft stability, and inflammation modulation [15]. Unlike flow cytometry, which provides aggregate data, immunohistochemistry allows researchers to study the spatial distribution and interactions of these cells within the tissue microenvironment. This capability could be particularly important in understanding the mechanisms that underpin successful graft retention and long-term integration.

Furthermore, histological analysis can evaluate the extracellular matrix (ECM), a crucial scaffold supporting graft retention. According to many researchers, the ECM not only stabilizes adipocytes, but also provides a framework for vascularization and cell proliferation [12]. Mechanical processing methods such as the cell-enriched lipotransfer (CELT) protocol prioritize the retention of ECM and SVF cells while removing mature adipocytes, resulting in improved graft quality and reduced adverse effects [16,17,18,19,20]. Furthermore, these findings suggest that the mechanical enrichment of stem cells does not substantially alter their secreted factors, making it a viable method for lipofilling applications without regulatory concerns.

Despite its potential, the lack of standardized protocols for histological evaluation presents a significant challenge to the field. Variations in staining methods, sample preparation, and reporting practices often lead to inconsistencies, limiting the comparability of findings across studies. For instance, the choice of histological stains—ranging from hematoxylin–eosin (H&E) for general tissue evaluation to specialized stains like Masson Trichrome and Sudan III for ECM and lipid analysis—can influence the interpretability of results. Although H&E is the gold standard for clinical routines, the reviewed staining approaches are primarily useful in research to evaluate functional and structural parameters [21,22]. Similarly, differences in sample preparation techniques, such as centrifugation, filtration, and mechanical processing, can affect the observed cellular and structural properties of the tissue [23,24]. While this underscores the primary rationale behind our emphasis on the selection of the preparation protocol, it should be noted that no correlation exists between the choice of staining and the preparation method of the fat tissue. Furthermore, challenges such as the presence of emulsified fat products, like nanofat, and artifacts introduced during histological preparation, must be carefully addressed to ensure accurate results [20,25].

The present scoping review aims to address these challenges by providing a comprehensive overview of the histological and immunohistochemical staining methods employed in adipose tissue research. The goal of this scoping review is not to generate new biological hypotheses, but rather to consolidate the existing methodological practices in adipose tissue histology and immunohistochemistry, thereby identifying patterns, gaps, and areas for improvement, and offering recommendations for standardizing histological evaluations. This review will serve as a valuable resource for researchers and clinicians, facilitating informed decision-making and providing a comprehensive overview of all pertinent subjects concerning the histology of adipose tissue.

## 2. Methods

### 2.1. Literature Search

This review adhered to the methodology outlined in the JBI Manual for Evidence Synthesis, specifically Chapter 11 on scoping reviews, and followed the PRISMA guidelines for systematic reviews and “PRISMA Extension for Scoping Reviews (PRISMA-ScR): Checklist and Explanation” [26]. A systematic search of the relevant literature was performed across major databases, including PubMed, Ovid, and the Cochrane Library, covering studies published from inception until December of 2024. In our opinion, those 3 databases fit the research question the best. The Boolean operators used for the search included terms such as “lipoaspirate”, “fat graft”, “gauze rolling”, “decantation”, “coleman fat”, “celt”, “nanofat”, “lipofilling”, or “human fat”, combined with the term “histol*” to capture all variations of histological terminology. This search strategy ensured the comprehensive inclusion of studies addressing histological and immunohistochemical methods in adipose tissue research.

### 2.2. Manual Reference Search

In addition to database searches, all reference lists from the identified studies were manually reviewed to identify any additional relevant articles that might have been overlooked. This manual search step aimed to ensure completeness and minimize the risk of missing pertinent studies.

### 2.3. Literature Selection

All search results were imported into Zotero 7.0.11 (2024) to manage citations and remove duplicate entries. Following this, two independent reviewers screened the titles and abstracts to assess their relevance to the inclusion criteria. If the abstract lacked sufficient detail to determine eligibility, the full-text article was retrieved for further evaluation. The inclusion criteria focused on studies utilizing histology or immunohistochemistry in undigested human adipose tissue or its processed derivatives, with a specific emphasis on peer-reviewed articles written in English and reporting on qualitative or quantitative tissue markers. Studies were excluded if they primarily focused on animal adipose tissue to retain the focus on the identification of specific features, isolated stem cells rather than whole tissue, or employed highly complex manipulations beyond standard experimental protocols. To avoid redundancy, duplicate patient cohorts were identified by cross-referencing study characteristics such as authors, institutions, and sample populations, ensuring that each dataset was only included once.

### 2.4. Data Extraction

Data extraction was performed using a standardized form developed specifically for this review. Two independent researchers extracted details such as study title, author, year of publication, study design, sample size, patient demographics, histological staining methods, qualitative markers (e.g., fibrosis, necrosis, inflammation), quantitative markers (e.g., SVF cell counts, mature adipocyte viability), and preparation protocols (e.g., centrifugation, filtration). The role of histologists or pathologists among the authors was also documented. Discrepancies during data extraction were resolved through discussion, with unresolved disagreements referred to a third reviewer for adjudication.

### 2.5. Statistical Analysis

This scoping review used descriptive–analytical methods, including frequency, percentage, and data charting using Microsoft Excel 2405 (2021) (Table 1). The screening process was evaluated by independent pairs of authors. The graphs and tables were designed using Microsoft Excel 2405 (2021). The registration number for this scoping review is osf.io/au9vn.

## 3. Results

### 3.1. Prisma Flow Diagram

The following flow diagram illustrates the process used to identify studies that meet the inclusion criteria, as outlined in PRISMA 2020 (Figure 1).

### 3.2. Study Selection

A total of 1122 articles were identified through database searches. After the removal of 428 duplicate entries, 694 articles were screened based on titles and abstracts. At total of 349 full-text articles were assessed for eligibility, of which 183 were excluded. Ultimately, 166 studies meeting the inclusion criteria were selected, representing 166 investigations that analyzed the histological and immunohistochemical techniques for adipose tissue evaluation (Table 2).

### 3.3. Patient and Animal Sample Characteristics

Across the studies, the mean number of human participants per study was 7.1 (±10.3), while the mean number of animals used for human transplanted fat tissue was 22.8 (±31.5), predominantly using immunodeficient mice. Human fat samples were predominantly harvested via liposuction from areas such as the abdomen, axilla, or flanks. We excluded all studies that featured only animals, but we included those that focused on the transplantation of human adipose tissue into animals.

### 3.4. Histological Staining Techniques

Hematoxylin–eosin (H&E) was the most frequently used stain, reported in 152 studies (91.5%). Specialized stains such as Masson Trichrome (14.5%) and Sudan III or Sudan Black (2.4%) were utilized to assess extracellular matrix (ECM) components and lipid structures, respectively. In five studies, only immunohistochemical techniques were employed without standard histological staining (Figure 2).

### 3.5. Immunohistochemical Analysis

Immunohistochemistry was performed in 55.4% of studies. The most used markers included CD31 and vWF for endothelial cells, CD34 and CD90 for stromal vascular fraction (SVF) cells, and perilipin for mature adipocytes. Additionally, markers such as CD68 and F4/80 were employed to identify inflammatory cells, while ECM components were evaluated in a subset of studies using collagen-specific antibodies (Figure 3).

### 3.6. Preparation Methods

Centrifugation, filtration, gauze rolling, decantation, and mechanical processing were the primary methods for preparing adipose tissue. Seventeen studies (15.1%) did not report all preparations in detail. In 33 studies, fresh lipoaspirates were analyzed, while 139 studies focused on explanted fat. Variations in preparation methods were reported to influence histological and immunohistochemical outcomes (Figure 4).

### 3.7. Qualitative Markers, Semiquantitative Analysis, and Quantitative Markers

Qualitative markers, including fibrosis, necrosis, inflammation, and oil cyst formation, were evaluated in 137 studies. Fibrosis was analyzed using ECM-targeted stains such as Masson Trichrome, while necrosis and oil cysts were evaluated using H&E or lipid-specific stains like Oil O Red. Inflammation was analyzed based on specified antibodies against immunocompetent cells as macrophages (Figure 5).

Of the 137 qualitative markers studied, 60 studies [8,9,24,25,29,40,42,44,45,49,52,56,58,65,67,69,70,76,85,86,87,88,90,95,97,98,99,100,101,103,105,106,108,112,114,124,125,127,134,135,136,142,146,147,150,151,152,155,156,159,162,163,165,171,174,175,176,177,178,180] utilized semiquantitative scoring systems, in most cases ranging from 0 to 5. An average of 2.5 ± 0.9 authors contributed to the evaluation process. They analyzed a mean of 3.0 ± 0.7 qualitative markers.

Quantitative markers, including SVF cell density, mature adipocyte counts, or vascular structures, were assessed in 156 studies. CD31 and vWF were used to quantify endothelial cells, while perilipin staining measured adipocyte integrity. ECM components were less frequently evaluated quantitatively, with only a minority of studies employing advanced image analysis for semi-automated quantification (Figure 6).

### 3.8. Author Expertise

Histological or pathological expertise was reported in 20 studies, with 12 studies listing such experts as first or last authors.

### 3.9. Reporting and Methodological Gaps

Twenty-five studies did not specify their preparation methods. Variations in sample preparation methods were observed across studies, as seen in Table 2. Additionally, the use of advanced imaging or analysis techniques was limited.

## 4. Discussion

This overview of stain-to-feature associations helps readers to understand the methods used and why they are selected for specific histological evaluations.

### 4.1. Staining in Histology

In the current context of bioscience, standard histology is a readily accessible resource for many research groups. The predominant staining method for diagnostic practice is in many cases hematoxylin–eosin. This method utilizes hemalaun to stain all basophilic/acidic structures, such as the nucleus, blue, and the synthetic eosin to stain all acidophilic/basic structures, such as the cytoplasm and collagen, red.

This study showed that more specialized stains than those used in clinical routine are commonly used for regenerative medicine and tissue engineering research.

HPS (haematoxylin–phloxine–saffron)/Masson Trichrome—An amendment to the standard H&E staining method is namely HPS (hematoxylin–phloxine–saffron) staining. Its objective is to enhance the contrast between connective tissue and cytoplasm. The transition from eosin to phloxine aims to enhance the contrast by producing a deeper red hue that is less susceptible to fading than that of eosin [184]. In contrast, saffron is employed to stain connective tissue in a yellow hue [185,186].

The Masson Trichrome staining method stains three colors. Collagen is stained blue to green, muscles are stained red, and nucleus are stained black [187]. This staining method is more commonly used than HPS in the evaluation of human fat in scientific research (twenty-four times versus three times). As it stains muscles as well, this method can be used to evaluate tissues that include them, which can occur in explants.

Sudan III/Oil “O” Red—The differentiation between connective tissue and mature adipocytes can be accomplished by Sudan III or Oil O Red staining. Sudan III staining produces an orange-to-red hue [188], while Oil O Red staining yields a red hue coloration [189]. The surface of the red-orange area can be analyzed with image processing software, allowing the success of the processing to be evaluated.

### 4.2. Staining in Immunohistochemistry

Stem cell markers—In immunohistology, it is standard practice to use a single antibody per histological slice, as all antibodies stain the target structure in a brown hue, making differentiation between them on the same slide challenging [147]. For EPCs, CD31 and the von Willebrand factor (vWF) are common markers: pericytes are identified using CD146, and ADSCs typically express CD34, CD71, and CD90 [15]. In situations where distinguishing ADSCs from hematopoietic stem cells is critical while maintaining the spatial resolution, immunofluorescence may be employed. For example, Zimmerlin et al. (2010) demonstrated that immunofluorescence allows for the simultaneous identification of two to three targets [190], and the inclusion of CD45 can help exclude hematopoietic lineage cells.

A further limitation is that, as illustrated in Figure 7, there are areas of overlap in the specialties of all targets, to varying degrees. This should be considered when designing a study.

If the differentiation of the three subsets was not of interest, CD90 was identified as a marker expressed in all three subsets at a high level (over 50% of described subsets), thus making it a SVF cell marker. Furthermore, ɑ-SMA is able to identify blood vessels, and, therefore angiogenesis [79].

Mature adipocyte markers—The structural integrity and viability of mature adipocytes can be evaluated based on the presence or absence of an intact perilipin layer [45,130,172]. It should be noted that a simple disruption of adipocytes can also be caused by fixation and cutting during the process of histological preparation. To avoid this methodological flaw, it is advisable to spare parts of the slide that are obviously distorted due to histological preparation [191].

Inflammatory cells in particular macrophages and their markers—This complication, which can occur in every transplantation of lipoaspirate, cannot be measured using the previously discussed methods [155]. The localization of immunocompetent cells, such as macrophages and other white blood cells, outside of blood vessels, within the tissue, is an indicator of inflammation [192]. Antibodies that are specific to, for example, macrophages, can therefore be used to identify areas of inflammatory activity. Two markers that fulfill this function are F4/80 and CD68 [40,149]. F4/80 is the murine equivalent to human EMR1, which is also expressed on eosinophile cells [193,194]. Since studies aiming to identify side effects in lipografts are primarily based on explants from human fat as xenografts, using this antibody is an appropriate methodology. CD68 is not murine but human-specific, while its murine ortholog is called macrosialin [195]. It is mainly stored inside macrophages and to a lesser extent on the cell surface. These analyses are mainly supported in scientific research and not in clinical routine.

Extracellular matrix (ECM) markers—While the ECM mainly consists of collagenase I–IV, fibronectin and laminin, components that are crucial for its characteristics, such as neovasculogenesis and insulin metabolism, have been studied [12,196]. Given the limitations of conventional histology in differentiating between these diverse components, immunohistochemistry offers a solution for achieving the spatial resolution of the ECM in fat tissue [197,198].

This scoping review provides a comprehensive overview of the histological and immunohistochemical methods employed in adipose tissue research, highlighting both the strengths and limitations of the current approaches. The dominance of H&E staining as the primary method of analysis reflects its broad applicability and accessibility, making it a staple in the evaluation of adipose tissue. Its ability to provide a general overview of tissue morphology makes it indispensable for assessing both qualitative markers, such as fibrosis, necrosis, and inflammation, and quantitative markers, including adipocyte integrity and SVF cell density. However, the reliance on H&E alone may obscure more nuanced details of tissue architecture, necessitating the integration of specialized stains for a more comprehensive analysis.

Specialized staining methods, such as Masson Trichrome and Sudan III and Sudan Black, demonstrated significant utility in evaluating the ECM and lipid components of adipose tissue, respectively. Despite their potential to enhance the resolution of histological evaluations, these methods remain underutilized, with only 22% of studies incorporating them. The underrepresentation of these techniques underscores a missed opportunity to gain deeper insights into the structural and functional properties of adipose tissue. Given the pivotal role of the ECM in graft retention and the pathophysiology of fibrosis, future research should prioritize the inclusion of these staining methods to address current gaps in the knowledge.

Immunohistochemistry has emerged as a critical tool for elucidating the cellular composition of adipose tissue, enabling the identification and localization of specific cell types within the tissue microenvironment. Markers such as CD31, vWF, CD34, and perilipin offer insights into vascularization, stem cell content, and adipocyte viability. However, the prevalent use of single-marker staining limits the ability to assess complex cellular interactions and their spatial dynamics. Multiparametric approaches, such as multiplex staining immunofluorescence, offer a promising solution by allowing the simultaneous analysis of multiple targets, thereby providing a more holistic view of tissue composition and behavior [147]. The limited adoption of such advanced techniques highlights a critical area for methodological improvement in future studies.

Differences in liposuction techniques, centrifugation protocols, and mechanical processing directly influence the observed histological features, as evidenced by the work of Condé-Green et al. [24]. Although the outcomes of the preparation parameters were not extracted systematically in this scoping review, the lack of detailed reporting on the preparation methods in 25 of the included studies, which can be found in Table 2, marked with “-” for “No comment”, further exacerbates this issue. Standardizing preparation protocols and ensuring comprehensive methodological reporting are essential steps toward enhancing the reliability and generalizability of histological findings in adipose tissue research.

The histological structure of human fat tissue may be influenced by multiple external factors, such as donor age, harvesting site, and pathologies like lipedema. However, in the majority of cases, these factors were either underreported or ambiguously reported. Had they been part of the exclusion criteria, this would have resulted in a significant reduction in the number of included articles, thereby compromising the synthesis. As a result, we chose not to extract these findings in an incomplete and misleading form, but rather to highlight this issue as a limitation. Readers should be aware that pooling the results means that the most common staining methods may not be optimal for answering specific scientific questions about specific pathologies, such as lipedema or metabolic disorders. This review only provides recommendations for cell- and structure-specific staining, not for staining specific to external factors. To enable better comparison and meta-analytic synthesis, we encourage future studies to report these variables more consistently. Additionally, the sample size of patients included in a study often varied considerably and was only reported when applicable. This should be considered when interpreting the frequency of reported staining.

The exclusion of studies utilizing whole tissue fat from animals was based on inter-species disparities, in order to maintain the clinical, human-specific relevance of this review. A significant difference in histological evaluation is the variability in adipocyte size and their distinct behavior in anabolic processes. As Börgeson et al. summarized, human adipocytes tend to be larger [199], and therefore their yield in a histological slide cannot be properly compared to that of mice. Furthermore, they report that, human adipocytes typically undergo hypertrophy, while in mice, a hyperplastic pattern is observed during periods of fat gain [199]. A further point to consider is the method of fat extraction. In the case of small mammals, the fat is collected as a whole fat pad, for example, from the inguinal region, and then minced with scissors [85]. In contrast, human fat is mostly obtained through liposuction. The investigation by Iyyanki et al. sought to ascertain the extent to which these methodologies impact the overall cell yield of SVFCs in white adipose tissue. Their findings revealed significant disparities between liposuction and the excision of fat. Furthermore, they have demonstrated that the harvesting site of WAT exhibits substantial variation in terms of SVFC yield, with the abdomen, the axilla, and the flanks being compared [200,201,202,203].

To our knowledge, this is the first scoping review examining commonly used staining methods and their applications in adipose tissue histology and immunohistochemistry. This review underscores the importance of collaboration with histologists and pathologists to enhance the quality and interpretability of histological analyses. While histological expertise was not a prerequisite for meaningful contributions, as evidenced by the diverse author affiliations in the included studies, involving experts in the field can improve the methodological rigor and depth of analysis. Furthermore, the technician and the laboratory equipment could influence the methodologies and results of histology. Therefore, in the future, a certification program for providers of histological techniques should be developed. Additionally, the adoption of advanced imaging and semi-automated analysis techniques can further enhance the resolution and accuracy of histological evaluations, providing new avenues for exploring the complexities of adipose tissue dynamics.

It is imperative to evaluate the laboratory possibilities in histology. It is important to note that not all laboratories are equipped to offer standard histology or immunohistochemistry. In circumstances where institutional resources are limited, or where the research group has no prior experience of histological assessment, it is advisable to collaborate with departments of histology or pathology for both consultation and execution. In addition to stains and antibodies, it is imperative to consider the adequacy of software for viewing slides, and image processing software or other digital tools for slide evaluation capable of the semi-automated analysis of target structures.

### 4.3. Recommendations and Operative Considerations

The timing of the histological assessment is crucial in determining the type and interpretive value of the resulting data. Pre-implantation analysis provides information on baseline tissue composition and cellular architecture, but it does not reflect in vivo behavior or integration. Conversely, post-explantation histology allows for the evaluation of inflammatory responses, fibrosis, vascularization, and tissue remodeling. However, variability related to biopsy site, timing, and preservation often affects such samples. These limitations underscore the importance of interpreting histological findings within the context of experimental design and sampling strategies. The recommended stainings for certain cells and complications can be found in Table 3.

Among the studies reviewed, 139 analyzed excised fat as whole tissue, whereas 33 examined lipoaspirates. Lipoaspirates typically present as a non-coherent mixture of mature adipocytes and SVF and ECM cells, whereas whole tissue explants retain a structured, coherent morphology with intact blood vessels. The histological assessment of lipoaspirate poses challenges, including the difficulty of evaluating oil cysts, fibrosis, and inflammation. Extracellular matrix accumulation may be artificially induced and indistinguishable from fibrosis, while extravasated immunocompetent cells may be mistaken for inflammation. The evaluation of oil cysts is further complicated by tissue scaffold disruption, which can mimic cystic structures. Therefore, qualitative markers in lipoaspirates are not typically assessed unless they have undergone implantation and subsequent retrieval, allowing for tissue integration and structural regeneration.

For explants, biopsies, or intact fat excisions, qualitative markers indicative of necrosis, fibrosis, oil cysts/vacuolation, and inflammation/cell infiltration can be effectively analyzed. These markers represent key histological side effects widely discussed in the literature. The most common evaluation approach is semiquantitative, employing a scoring system from 0 to 5, where 0 denotes no evidence of the specified marker and 5 indicates maximal presence. This method facilitates interstudy comparability and minimizes subjectivity, particularly when assessments involve multiple authors.

In the context of evaluating lipoaspirate samples, a quantitative evaluation can also prove beneficial. It is also recommended that this quantitative evaluation be performed on whole tissues. The structures of interest are mainly SVFCs, mature adipocytes, and the ECM. The staining of the ECM is analogous to that of fibrosis.

SVF cells hold the capacity to differentiate into a multitude of tissues, including bone, adipose tissue, neural cells, and cartilage [204,205]. Given the heterogeneous nature of the SVF, we focus on three key subsets of SVF cells: adipose-derived stem cells (ADSCs), endothelial progenitor cells (EPCs), and pericytes [206]. The aspects of interest are the generation of new adult cells, the fact that inflammatory processes are suppressed, and that neovascularization occurs [36,169,180]. The first point is since adult adipocytes are more susceptible to shear force, which arises when fat is applied through small-diameter cannulas [20]. Moreover, the absence of replacement by stem cells would result in volume loss at the graft site. The destruction of adult adipocytes during application releases free lipids, which can cause inflammation, while SVF cells can act locally and in an anti-inflammatory manner [207]. Neovascularization is of interest because in fresh lipoaspirate, there are insufficient functioning vessels, as they are ripped out of place and into pieces by the liposuction cannula [58]. To ensure long-term graft survival, sufficient vessels are constructed initially, from the EPCs and pericytes, to secure continuous blood flow [71].

Fibrosis is characterized by the uncontrolled expansion of a dermal and cell-poor ECM, best evaluated using Masson Trichrome or hematoxylin–eosin–saffron staining. Inflammation and cell infiltration can be detected via H&E staining, though immunostains for macrophages such as CD68 and F4/80, providing greater accuracy. Necrosis, most effectively analyzed through H&E staining, can also be assessed using hypoxia-inducible factor-1 alpha (HIF-1α) antibodies to detect oxygen deficiency. Oil cysts and vacuoles require precise diameter measurement and lipid-specific staining methods, such as Oil O Red or Sudan stains, to distinguish them from adipocytes. SVF cells, depending on the research focus, can be analyzed using H&E staining for general evaluation or immunohistochemistry, including FACS and flow cytometry, for precise cell subset identification. Mature adipocytes are best evaluated via H&E staining and perilipin immunostaining, with mice to human differentiation achieved using vimentin immunostaining.

From a methodological perspective, it is imperative to meticulously document the preparation of the fat tissue to ensure maximum comparability with existing studies. Moreover, the publication of the obtained histological slides in their entirety is advantageous, as it enables other researchers to provide a more comprehensive overview than a detailed description of histological slides alone.

## 5. Conclusions

This scoping review highlights the histological and immunohistochemical techniques commonly employed for adipose tissue analysis. Hematoxylin–eosin remains the most prevalent stain, while markers such as CD31, CD34, and perilipin are frequently used to assess key cellular subpopulations. Variability in methodologies—including preparation techniques and marker selection—underscores the need for standardized protocols to enhance consistency and comparability across studies.

Future research should prioritize the standardized documentation of sample preparation, comprehensive marker panels, and the adoption of advanced imaging techniques. Collaborations with histology and pathology experts are highly recommended to improve methodological rigor. These efforts will enable more robust and reproducible findings, driving advancements in adipose tissue-based therapies and regenerative medicine applications. The findings of this review are a valuable reference for researchers planning a histological evaluation of adipose tissue in the field of regenerative medicine and tissue engineering. They also support the idea of improving methodological consistency in clinical and translational contexts.

## Figures and Tables

**Figure 1 cells-14-00898-f001:**
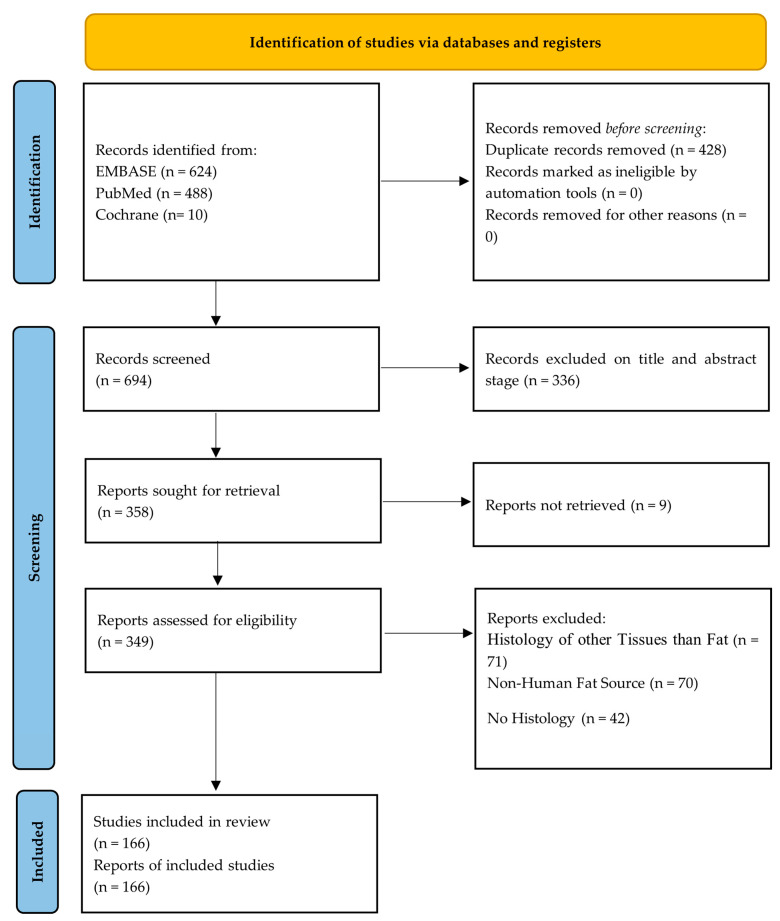
PRISMA 2020 flow diagram to identify the studies that fulfill the inclusion criteria [27].

**Figure 2 cells-14-00898-f002:**
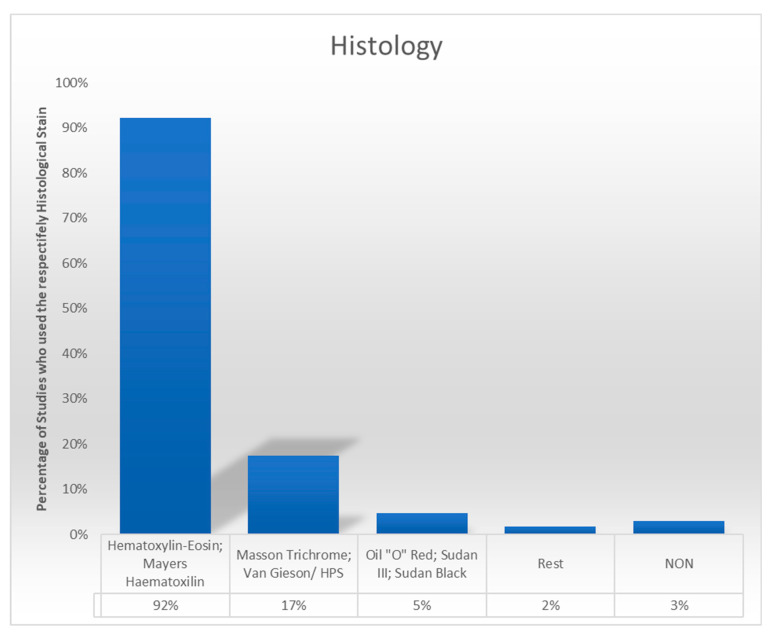
The column diagram demonstrates the frequency with which a specific staining method was employed. It should be noted that the sum of the percentages in this diagram does not equal 100% due to the utilization of multiple staining methods in numerous cases. Fifty-two percent of studies combined histological and immunohistochemical techniques to analyze both cellular and structural components of adipose tissue. The integration of these methods allowed for a comprehensive evaluation of tissue characteristics, though methodological variations were noted among studies.

**Figure 3 cells-14-00898-f003:**
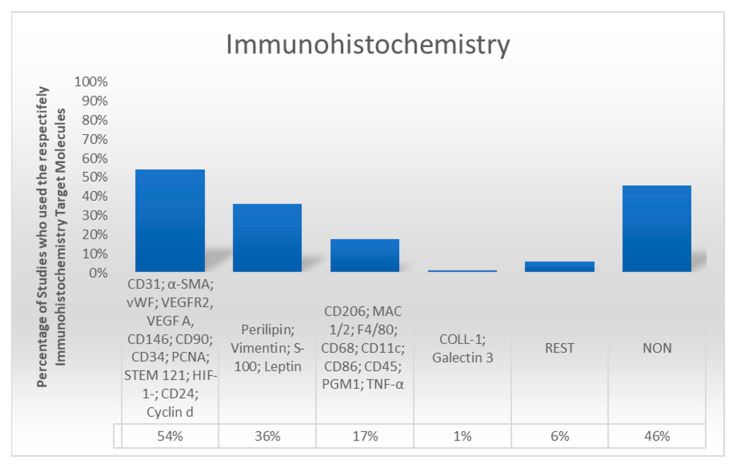
The column diagram demonstrates the frequency with which a specific antibody was used for immunohistochemistry. Single-marker immunohistochemistry was used in 33 studies and multiparametric techniques 57 times, while in 76 studies, no immunohistochemistry was assessed.

**Figure 4 cells-14-00898-f004:**
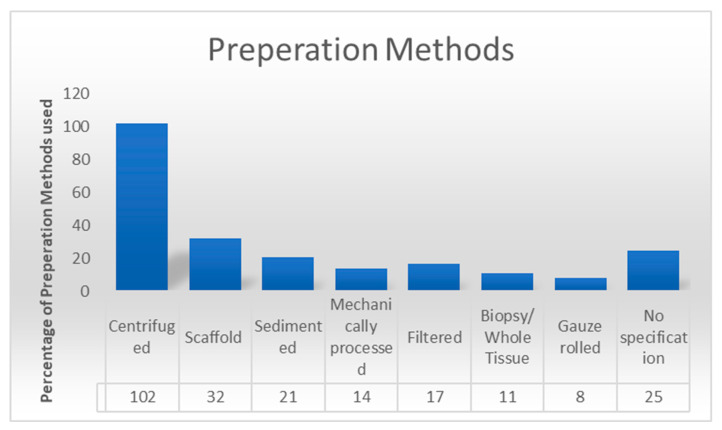
The column diagram demonstrates the frequency with which a processing method was employed.

**Figure 5 cells-14-00898-f005:**
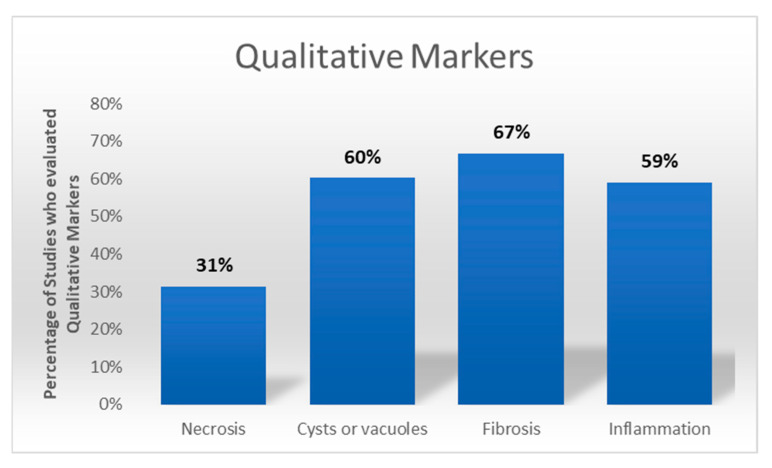
The graph illustrates the frequency with which the corresponding qualitative characteristics were evaluated among the studies (166).

**Figure 6 cells-14-00898-f006:**
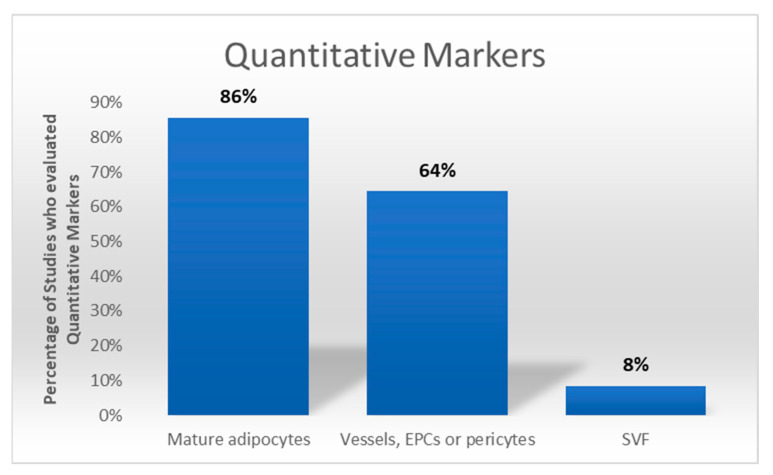
The graph illustrates the frequency with which the corresponding quantitative characteristics were evaluated among the studies (166).

**Figure 7 cells-14-00898-f007:**
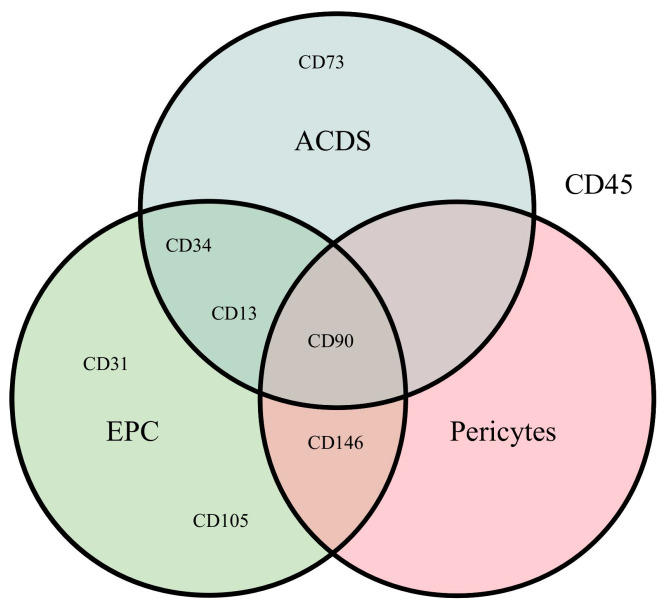
The graphic illustrates which markers are present in at least 50% of the circles they are included in, according to Thitilertdecha et al. (2020) [15]. The yellow circle represents ADSCs, the magenta circle represents pericytes, and the green circle represents EPCs. It should be noted that CD45 is outside all three circles, as it is specific to hematopoietic lineage cells. Additionally, a considerable number of markers are present in more than 50% of a subgroup in more than one subgroup, including CD13, CD34, CD90, and CD146.

**Table 1 cells-14-00898-t001:** Inclusion and exclusion criteria of this scoping review.

Criteria	Inclusion	Exclusion
Study Type	In vitro and in vivo studies with histology of fat tissue	Case reports, editorials, commentaries, reviews without original data, systematic reviews, or meta-analyses
Population	All human fat studies including animals transplanted with human fat	All animal fat studies
Intervention	Whole tissue fat	Studies focusing only on stem cells of the fat (SVF) and their histology or complex manipulated fat
Publication Date	All studies available (1972–2024)	-
Language	Published in English	Non-English language studies, unless translated versions are available
Peer-reviewed	Studies must be published in peer-reviewed journals	Non-peer-reviewed articles, conference abstracts, or unpublished theses

**Table 2 cells-14-00898-t002:** The factors regarding the evaluation of quality and the staining protocols used in the included studies. C: Centrifugation; S: Sedimented; M: Mechanically processed; Sc: Scaffold *; F: Filtrated or washed; G: Gauze rolled; B: Biopsy; -: No comment **; MAs: Mature adipocytes; SVFCs: Stromal vascular fraction cells; V: Vessels or vascular progenitor cells; I: Inflammation; Cy: Cysts or vacuoles; N: Necrosis; Fi: Fibrosis; and CD: Cluster of differentiation. (i.e., * scaffold means all whole tissue fat derivatives that were mixed with ADSCs or SVFCs and all synthetic materials mixed with whole tissue fat. ** No comment means that the methodology employed in the preparation of the fat tissue was not documented or comprehensible.).

Autor/Year of Publication	Tissue	Analyzed Attributes	Staining Methods
Adanali et al., 2002 [28]	C	MAs	Sudan Black
Adem et al., 2022 [29]	S	V, I, Cy, Fi	HE, Anti-CD31
Afanas’eva et al., 2018 [30]	-	MAs, V	HE
Agostini et al., 2012 [31]	C	MAs, N, Fi	HE, Sudan Black
Ansorge et al., 2014 [32]	C, S, F	MAs, V, I, Cy, Fi	HE
Atanassova et al., 2001 [33]	B	MAs, SVFCs	Sudan III, Anti-S-100 (S-100-protein)
Bach-Mortensen et al., 1976 [34]	B	MAs, V, I, Fi	Toluidine blue
Bae et al., 2015 [35]	C, Sc	MAs, V, I, Fi	HE, Anti-CD31
Baker et al., 2009 [36]	F	MAs, V	Anti-CD31
Bauer et al., 1995 [37]	B	MAs, V, I, Fi	HE
Bellas et al., 2013 [38]	Sc	MAs, V, I	HE, MT, Oil O Red
Bi et al., 2024 [39]	C	MAs, V, I, Cy, N, Fi	HE, MT, Anti-perilipin, Anti-CD31, Anti-CD206, Anti-MAC2 (Galectin-3)
Borrelli et al., 2020 [40]	S, Sc	MAs, V, I, Cy, N, Fi	HE, MT, Anti-perilipin, Anti-CD31, Anti-F4/80 (EGF-like module containing, mucin-like, hormone receptor-like sequence 1)
Bryant et al., 1983 [41]	B	MAs, V, Fi	HE
Chai et al., 2023 [42]	-	MAs, V, I, Cy, N, Fi	HE, Anti-perilipin, Anti-CD31, Anti-Ki67 (Marker of Proliferation Ki-67)
Chajchir and Benzaquen et al., 1989 [43]	-	MAs, V, Cy, Fi	HE
Chen et al., 2021 [44]	-	MAs, V, I, Cy, Fi	HE, Anti-CD34, Anti-Ki67, Anti-VEGF (Vascular Endothelial Growth Factor)
Chen et al., 2022 [45]	S, F	MAs, V, I, Cy, Fi	HE, Anti-perilipin, Anti-CD31
Chen et al., 2022 [46]	Sc	MAs, V	HE, Anti-CD31
Chen et al., 2024 [47]	F	MAs, I, Cy, Fi	HE, Anti-perilipin, Anti-caspase3
Chia et al., 2015 [48]	C, F	MAs, N	HE
Chung et al., 2019 [49]	C	MAs, V, I, Cy, Fi	HE, Anti-CD31
Cicione et al., 2016 [50]	C	MAs, SVFCs	HE, Sudan III
Cicione et al., 2023 [51]	M, -	MAs, SVFCs	HE, Anti-PCNA (Proliferating Cell Nuclear Antigen), Anti-CD31, Anti-CD34, Anti-COLL1 (collagen)
Condé-Green et al., 2010 [24]	C, S	MAs, N	HE, PAS-Reaction
Condé-Green et al., 2010 [23]	C, S	MAs	HE, PAS-Reaction
Condé-Green et al., 2013 [52]	C, S, Sc	MAs, V, I, Cy, N, Fi	HE
Craft et al., 2009 [53]	M	MAs, I, Cy, Fi	MT, Anti-S-100
Cui and Pu et al., 2009 [54]	C	MAs, N	HE
Cui and Pu et al., 2010 [55]	C	MAs, N, Fi	HE
Davis et al., 2013 [56]	C	MAs, Fi	HE
Debald et al., 2017 [57]	F	V, N, Fi	HE
Deleon et al., 2021 [58]	S	MAs, V, I, Cy, Fi	HE, Anti-CD31, Anti-perilipin
Dimitroulis et al., 2011 [59]	-	MAs, Cy, N	HE
Dobran et al., 2017 [60]	-	Fi	HE
Dong et al., 2022 [61]	Sc, G	MAs, V, I, N, Fi	HE, Anti-CD31, Anti-perilipin, Anti-STEM 121 (Microtubule-associated protein 1 light chain 3 alpha)
Eigenberger et al., 2022 [20]	C, S, M	MAs, SVFCs, V	HE
Eskalen et al., 2024 [62]	C	MAs	HE, Anti-perilipin
Fan et al., 2023 [63]	C, M	MAs, V, Fi	HE, Anti-perilipin, Anti-CD31
Ferguson et al., 2008 [64]	C, F	MAs, N	HE
Filson et al., 2016 [65]	C	V, I, Cy, Fi	HE
Fisher et al., 2013 [66]	C, F, G	V	Anti-CD31
Genç et al., 2022 [9]	S	MAs, I, Cy, N, Fi	HE, MT, Anti-perilipin
Girard et al., 2015 [67]	C, S	MAs, V, Cy, N, Fi	HE, MT
Ha et al., 2015 [68]	C, Sc	Cy	HE
Hamed et al., 2010 [69]	-	MAs, V, I, Cy, Fi	HE, Anti-CD31, Anti-CD68, Anti-VEGF, Anti-EPOR (erythropoietin receptor)
Hamed et al., 2012 [70]	Sc, -	MAs, V, I, Cy, Fi	HE, Mayer’s hematoxylin
Harris et al., 2019 [71]	Sc, -	MAs, V, I, Cy, N, Fi	HE, Anti-CD31
He et al., 2019 [72]	Sc, -	MAs, V, I, Fi	HE, Anti-CD31, Anti-MAC2, Sirius Red
He et al., 2023 [73]	C, M	MAs, V, Cy	HE, Anti-CD31, Anti-Ki67
Herly et al., 2017 [74]	-	MAs, I, Cy, N, Fi	HE, van Gieson, Anti-CD68, Anti-PGM1 (phosphoglucomutase 1), Anti-S-100
Hersant et al., 2018 [75]	C	MAs, V, I, Fi	HE, Anti-perilipin, Anti-CD31, Anti-CD45
Hivernaud et al., 2017 [76]	C, S, F	I, Cy, Fi	HPS
Ho et al., 2022 [77]	C	MAs, I, Cy, Fi	HE, Anti-perilipin, Anti-F4/80, Anti-caspase3
Hoareau et al., 2013 [78]	C, S	MAs, Fi	HES, MT
Hsiao et al., 2021 [79]	C	MAs, V, I, Cy, Fi	HE, Anti-perilipin, Anti-CD31, Anti-αSMA (Alpha-Smooth Muscle Actin)
Hu et al., 2018 [80]	C	MAs, V, Cy, Fi	HE, Anti-CD31
Huang et al., 2017 [81]	S, Sc	MAs, V, Cy, Fi	HE, MT, Anti-CD31
Ichikawa et al., 2005 [82]	B	MAs, N	HE
Janarthanan et al., 2023 [83]	C	MAs, V, I, Cy, N	HE, Anti-CD31, Anti-perilipin, Anti-vimentin
Jia et al., 2024 [8]	F	MAs, V, I, Cy, N, Fi	HE, Anti-perilipin, Anti-CD31
Jiang et al., 2015 [84]	C, S, M, Sc, F, G, B, -	V, Fi	HE
Jiang et al., 2023 [85]	S	MAs, V, I, Cy, N, Fi	HE, MT, Anti-perilipin, Anti-CD31, Anti-CD206
Jin et al., 2021 [86]	G	MAs, V, I, Cy, N, Fi	HE, Anti-perilipin, Anti-CD31
Jung et al., 2014 [87]	C	MAs, V, I, Cy, Fi	HE, Anti-CD31
Kakudo et al., 2013 [88]	Sc	MAs, V, I, Cy, Fi	HE, Anti-vWF (von Willebrand factor)
Kamel et al., 2014 [89]	C, F	MAs	HE
Kanamori et al., 2001 [90]	B	MAs, V, Cy, Fi	HE
Kelmendi-Doko, 2017 [91]	C	V	HE, Anti-CD31
Kelmendi-Doko et al., 2014 [92]	C	V	HE, Anti-CD31
Khater et al., 2009 [93]	C	MAs, SVFCs, V, I, Cy, N	HE, Anti-cyclin D1, Anti-leptin
Kijima et al., 2012 [94]	B	MAs, N, Fi	HE, Oil O Red
Kim et al., 2018 [95]	C	MAs, V, I, Cy, Fi	HE, Anti-perilipin, Anti-VEGF
Kim et al., 2022 [96]	C	MAs, V	HE, Anti-CD31, Anti-perilipin
Kim et al., 2023 [97]	C	MAs, V, I, Cy, Fi	HE
Kim et al., 2024 [98]	C, Sc	MAs, V, I, Fi	HE, MT, Anti-perilipin, Anti-CD31
Kirkham et al., 2012 [99]	C	MAs, I, Fi	HE
Ko et al., 2011 [100]	C, Sc	MAs, I, Cy, N, Fi	HE
Kokai et al., 2017 [101]	C	MAs, I, Cy, Fi	HE, MT, Anti-perilipin, Anti-F4/80,
Kuwahara et al., 2003 [102]	-, Laser	MAs, V	HE
Lee et al., 2013 [103]	C	MAs, I, Cy, Fi	HE
Lei and Xiao et al., 2020 [104]	C	MAs, V, Cy, N, Fi	HE
Li et al., 2012 [105]	C	MAs, I, Cy, Fi	HE
Li et al., 2013 [106]	C	MAs, V, I, Cy, Fi	HE
Li et al., 2014 [14]	Sc, -	MAs, V, I, N, Fi	HE, Anti-CD31
Li et al., 2015 [107]	F	MAs, V, I, Cy, Fi	HE
Li et al., 2020 [108]	F	MAs, SVFCs, V, I, Cy, N, Fi	HE, Anti-perilipin, Anti-CD31, Anti-CD206
Li et al., 2022 [109]	C	MAs, I, Cy	HE, Anti-perilipin
Li et al., 2024 [110]	C, Sc	SVFCs, V, I, Cy, Fi	HE, Anti-perilipin, Anti-CD31, Anti-VEGF
Li et al., 2024 [111]	C, Sc	MAs, V, I, Cy, Fi	HE, Anti-perilipin, Anti-CD31, Anti-vimentin, Anti-CD68
Li et al., 2024 [112]	C	MAs, V, I, Cy, Fi	HE, Anti-perilipin, Anti-vWF
Liu et al., 2024 [113]	G	MAs, V, I, Cy, N, Fi	HE, MT, Anti-perilipin, Anti-CD31
Loder et al., 2023 [114]	C	MAs, V, I, Cy, Fi	HE, Anti-CD31, Anti-perilipin, Anti-HIF-1-α (Hypoxia Inducible Factor 1), Sirius Red
Lu et al., 2009 [115]	Sc, -	MAs, V, N, Fi	HE, Anti-CD31
Luan et al., 2017 [116]	C	I, Cy, Fi	HE
Luo et al., 2015 [117]	Sc, -	MAs, V, Cy, Fi	HE, Anti-CD31, Anti-vWF, Anti-VEGF
Lv et al., 2019 [118]	C	MAs, V, I, N	HE, Anti-perilipin, Anti-vWF, Anti-CD68
Major et al., 2023 [119]	-	MAs, V, Cy, Fi	HE, Anti-perilipin, Anti-CD31
Martin-Ferrer et al., 1989 [120]	-	Fi	HE
Massiah et al., 2021 [121]	C	MAs, V	HE, Anti-perilipin, Anti-Ki67
Mecott et al., 2022 [122]	-	MAs	Fluorescence microscopy
Medina et al., 2009 [123]	C	MAs, I, Cy, Fi	HE
Medina et al., 2011 [124]	C	I, Cy, Fi	HE
Merrifield et al., 2018 [11]	C	MAs, SVFCs, V, I, Cy	HE, Anti-perilipin, Anti-CD34, Anti-CD24, Anti-CD68, Anti-Ki67
Minn et al., 2010 [125]	C, F, G	V, I, N	HE
Mojallal et al., 2011 [126]	C, Sc	MAs, V, Cy, Fi	HPS, Oil O Red, Anti-vimentin
Nelissen et al., 2023 [127]	C, S	MAs, I, Cy, Fi	HE, MT
Nguyen et al., 2012 [128]	C	MAs, V, N	HES, Anti-CD31
Nicoli et al., 2014 [129]	S	MAs	HE, MT
Nie et al., 2023 [130]	-	MAs, I, Cy, Fi	HE, MT, Anti-perilipin
Nie et al., 2024 [131]	C	MAs, V, I, N	HE, Anti-perilipin, Anti-CD31, Anti-CD206, Anti-Ki67, Anti-CD86
Niechajev and Śevćuk et al., 1994 [132]	F	MAs, I, Fi	HE
Niță et al., 2013 [133]	C	MAs, SVFCs, V, I, Cy, Fi	HE, Anti-DLK1(delta-like 1 homolog)
Oh et al., 2011 [134]	C	MAs, V, I, Cy, Fi	HE, Anti-CD31
Olenczak et al., 2017 [135]	C, Sc	MAs, I, Cy, Fi	HE
Paik et al., 2015 [136]	C, Sc	MAs, V, I, Cy, Fi	HE, Anti-CD31
Palumbo et al., 2015 [137]	C, S	MAs	HE, Anti-vimentin
Park et al., 2013 [138]	B	I, Cy, N, Fi	HE
Park et al., 2017 [139]	C, Sc	V, I, Cy, N, Fi	HE, MT
Pelosi et al., 2017 [140]	B	MAs, V	Anti-perilipin
Philips et al., 2013 [141]	C	MAs, V, Cy, Fi	HE, Anti-CD31
Por et al., 2009 [142]	C	MAs, V, I, Cy, N, Fi	Oil O Red
Pu et al., 2008 [143]	C	MAs, N	HE
Pu et al., 2010 [144]	C	MAs, N	HE
Ragni et al., 2022 [145]	C, M	SVFCs, V	HE, Anti-CD31, Anti-CD90, Anti-CD146
Ramon et al., 2005 [146]	C, F	MAs, V, I, Cy, Fi	HE
Reddy et al., 2016 [147]	C	MAs, V, I, Cy, Fi	HE, Anti-perilipin, Anti-CD31, Anti-vimentin
Säljö et al., 2020 [10]	M, Sc	MAs, SVFCs, V, Fi	HE, Anti-CD31, Anti-CD90
Sesé et al., 2020 [148]	C, M	MAs, SVFCs	HE, MT
Sheng et al., 2022 [149]	C, G	MAs, V, I, Cy, N, Fi	HE, Anti-CD31, Anti-CD68
Shoshani et al., 2000 [150]	C	I, Cy, Fi	HE
Shoshani et al., 2001 [151]	C	MAs, I, Cy, Fi	HE
Shoshani et al., 2005 [152]	C	MAs, I, Cy, Fi	HE
Skorobac Asanin and Sopta et al., 2017 [153]	C	MAs, V	HE
Smith et al., 2006 [154]	C	MAs, I, Cy, N, Fi	HE
Sun et al., 2023 [155]	C	MAs, I, Cy, Fi	HE, Anti-perilipin, Anti-F4/80, Anti-CD206, Anti-MAC2, Anti-CD11c
Szychta et al., 2024 [156]	C	MAs, V, I, Cy, N, Fi	HE
Teng et al., 2014 [157]	C	MAs, V, I, Cy, N, Fi	HE
Tong et al., 2018 [158]	G	Cy	HE
Tran et al., 2024 [25]	C, M	MAs, V, I, Cy, Fi	HE, Anti-perilipin, Anti-CD31, Anti-F4/80
Ullmann et al., 2006 [159]	C	MAs, V, I, Cy, N, Fi	HE
Van Dongen et al., 2020 [160]	C, M	MAs, SVFCs, V	MT, Anti-CD31, Anti-αSMA
Von Heimburg and Pallua et al., 2001 [161]	S	MAs, I, Cy, Fi	HE
Wang et al., 2015 [162]	F	MAs, V, I, Cy, N, Fi	HE
Wei et al., 2019 [163]	C	MAs, V, I, Cy, N, Fi	HE, Anti-perilipin, Anti-CD31
Weisz and Gal, 1986 [164]	B	MAs, V	HE
Wu et al., 2020 [165]	C	MAs, V, I, Cy	HE, Anti-perilipin, Anti-CD31
Wu et al., 2023 [166]	C, M	MAs, V, I, Cy	HE, Anti-perilipin, Anti-CD31, Anti-MAC2, Anti-CD206
Xia et al., 2021 [167]	Sc, -	MAs, V, Cy, N	HE, Anti-perilipin, Anti-CD31, Anti-UCP1 (uncoupling protein 1)
Xie et al., 2010 [168]	C	MAs	HE
Xiong et al., 2018 [169]	C	V, Fi	HE
Xu et al., 2014 [170]	-	MAs, V, I, N, Fi	HE
Yagima Odo et al., 2007 [171]	B	I, N, Fi	HE
Yang et al., 2021 [172]	C, M, G	MAs	HE, Anti-perilipin
Yang et al., 2023 [173]	Sc, -	V, I, Cy, Fi	HE, MT, Anti-CD31, Anti-αSMA, Anti-HIF-1α, Anti-VEGF
Yang et al., 2024 [174]	C, Sc	MAs, V, I, Cy, Fi	HE, MT, Anti-perilipin, Anti-CD31, Anti-αSMA, Anti-MAC2
Yi et al., 2006 [175]	C	MAs, V, I, Cy, Fi	HE, Anti-vWF
Yi et al., 2007 [176]	C	MAs, V, I, Cy, Fi	HE, Anti-vWF
Yu et al., 2018 [177]	S, M	MAs, V, I, Cy, Fi	HE, MT, Anti-CD31
Yu et al., 2020 [178]	C	MAs, V, I, Cy, Fi	HE, MT, Anti-perilipin, Anti- αSMA
Yu et al., 2020 [13]	C, Sc	MAs, V, Cy, N	HE, von Kossa, Anti-perilipin, Anti-CD31, Anti-αSMA, Anti-Ki67, Anti-VEGF, Anti-ANA (antinuclear), Anti-TNFα (tumor necrosis factor α)
Yu et al., 2021 [179]	C	MAs, V, I, Cy, Fi	HE, MT, Anti-perilipin, Anti- αSMA
Zhan et al., 2017 [180]	Sc, -	MAs, V	HE, Anti-perilipin, Anti-CD31
Zhang et al., 2018 [181]	C, M	MAs, SVFCs, V, I, Cy, N, Fi	HE, MT, Anti-perilipin, Anti-CD206, Anti-MAC2, Anti-HLA
Zhao et al., 2023 [182]	M, Sc	MAs	HE
Zhu et al., 2021 [183]	C, M	MAs, I, Cy, Fi	HE, Anti-perilipin

**Table 3 cells-14-00898-t003:** Summary of all the recommended stainings for certain cellular equivalents or complications (SVF: stromal vascular fraction; MAs: mature adipocytes; ECM: extracellular matrix; H&E: hematoxylin eosin; and IHC: immunohistochemistry).

Tissue Preparation	Preservation of Architecture	Evaluable Histological Targets	Recommended Stains	Specific Characteristics/Limitations
Lipoaspirate	Low	SVF	H&E, IHC	Fragmented, non-coherent mixture; vessel and ECM structures disrupted; interpretation of inflammation and fibrosis unreliable
		MAs	Perilipin, Oil O Red	
		ECM	Masson Trichrome	
		Vascularization/Vessel Integrity	Anti-CD31, Anti-αSMA	Enables analysis of neovascularization and graft viability
Surgically Excised Fat (including explants)	High	SVF, MAs, ECM, and Vessels	See above	The same reasons are applicable for surgically excised fat as for lipoaspirate
		Fibrosis	Masson Trichrome, H&E-Saffron	Enables semiquantitative scoring (e.g., 0–5 scale)
		Inflammation/Immune Cell Infiltration	H&E, Anti-CD68, Anti-F4/80	
		Oil Cysts/Vacuoles	Oil O Red, Sudan stains	
		Necrosis	H&E, Anti-HIF-1α	

## Data Availability

The data is available in Table 1 and Table 2 and upon request from our authors.

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
