# Peer review of "Histology and Immunohistochemistry of Adipose Tissue: A Scoping Review on Staining Methods and Their Informative Value"

_cells, 2025, doi:10.3390/cells14120898_

Round 1
Reviewer 1 Report
Comments and Suggestions for Authors
The manuscript intitled “Histology and Immunohistochemistry of Adipose Tissue: A Scoping Review on Staining Methods and Their Informative Value” presents a well-structured scoping review of histological and immunohistochemical (IHC) staining methods used in the assessment of human undigested adipose tissue and its derivatives. The authors systematically reviewed 166 studies, identifying commonly used stains (e.g., H&E, Masson’s Trichrome), immunomarkers (e.g., CD31, CD34, perilipin), and methodological variability. They offer practical recommendations for improved standardization and reproducibility. The topic is potentially relevant to researchers and pathologists working in regenerative medicine, fat grafting, and translational histology.
The manuscript adheres to PRISMA and JBI methodology and highlights the need for better consistency in histological techniques applied to adipose tissue research. It adds value by not only reporting what stains are used but also by reflecting on their functional and interpretative implications. However, the manuscript does not clearly distinguish between diagnostic and research-driven applications of histological and IHC techniques. Several of the methods discussed—while useful for research quantification and mechanistic exploration—are not standard practice in diagnostic histopathology. Furthermore, the manuscript could benefit from a deeper critique of the limitations inherent to sampling and processing adipose tissue for histology, particularly regarding timing (pre- vs. post-inoculation) and the need to preserve tissue viability.
I - Specific Comments
1. Strengths and Weaknesses in Reported Data
The authors highlight the widespread use of H&E and commonly used immunomarkers, and they acknowledge methodological inconsistencies across studies. However, weaknesses such as incomplete reporting, lack of standard preparation protocols, and low use of advanced imaging are only descriptively addressed. A structured assessment of methodological quality (e.g., using a formal appraisal tool) would strengthen the analysis.
2. Identification of Gaps and Proposals for Alternatives
The authors effectively identify the absence of standardized staining protocols and inconsistent reporting as critical gaps. They offer constructive suggestions, such as adopting semiquantitative scoring systems and encouraging collaboration with pathologists. Proposals for increased use of digital tools and multiplex staining are appropriate and forward-looking.
3. Interpretation of Technique Frequency and Justification
While the authors provide a helpful quantitative overview of the most frequently used histological and immunohistochemical techniques, the rationale behind the selection of these techniques in the primary studies is not systematically explored. It remains unclear whether the prevalence of certain stains or markers (e.g., H&E, Masson’s Trichrome, CD31, CD34) reflects their technical suitability, accessibility, historical use, or methodological advantages.
Without this context, it is difficult to interpret whether frequently used techniques are also the most appropriate for the intended histological evaluation. To strengthen the interpretive value of the review, the authors should include a critical discussion on why these techniques might be preferred, supported by methodological or functional justifications where available. A comparative table linking techniques to their intended purpose, strengths, and rationale (when reported) would enhance the transparency and utility of the findings for researchers selecting appropriate tools for future studies.
4. Clarification on Diagnostic vs. Research Applications
While the manuscript provides a comprehensive overview of staining techniques used in adipose tissue research, it would benefit from a clearer distinction between stains required for diagnostic histopathology and those employed in experimental or research contexts. Hematoxylin-eosin (H&E) remains the gold standard for evaluating adipose tissue morphology, inflammatory infiltrates, and structural lesions in clinical diagnostic practice. Experienced pathologists can readily identify adipocytes, fibrous tissue, necrosis, and inflammatory cells using H&E alone. Thus, complementary stains like Masson’s Trichrome, Oil Red O, or immunohistochemical markers such as CD68 or F4/80 are not generally required unless a specific diagnostic challenge arises.
In contrast, in regenerative medicine and adipose tissue engineering, additional staining techniques serve an investigative rather than diagnostic purpose. Masson’s Trichrome is used to quantify fibrosis; immunomarkers allow phenotyping of SVF or immune cells; and lipid stains help visualize oil cysts or damaged adipocytes. These methods are essential for quantitative and mechanistic insights and should be clearly contextualized as such. We recommend that the authors clarify these differences to avoid conflating clinical and experimental goals.
5. Considerations on Timing and Impact of Histological Assessment
Another aspect deserving further elaboration is the timing of histological/IHC assessment and its impact on the types of data that can be retrieved.
Pre-inoculation analysis is valuable for evaluating graft quality and cell composition but requires sacrificing part of the material—making it suitable only for preclinical studies or tissue processing validation. This form of analysis informs the baseline characteristics of the graft but must be interpreted with care, as processing can introduce variability.
Post-inoculation assessment is more common in animal studies or when clinical tissue is re-biopsied. It allows analysis of graft integration, inflammatory responses, fibrosis, and cell survival—critical for interpreting treatment outcomes. However, the sampling time and method can affect tissue representativeness and comparability.
Moreover, destructive histological processing inherently compromises cell viability. To mitigate this, researchers often split the graft into aliquots (for histology vs. implantation) or rely on surrogate measures such as flow cytometry, metabolic assays, or non-destructive imaging (e.g., confocal microscopy).
We suggest the authors expand their discussion to acknowledge these limitations, as they directly shape the kind of histological evidence available in the literature and help readers understand why certain approaches are over- or under-represented.
6. Influence of External Factors and Population Heterogeneity
While the authors have limited their review to human studies and attempted to categorize adipose tissue by preparation type (e.g., lipoaspirate, microfat), the influence of external factors such as donor age, anatomical site, pathological status, or tissue processing method is not fully accounted for. These factors can significantly impact histological architecture, marker expression, and immune cell infiltration — yet appear to be unevenly reported across the included studies.
Moreover, some summaries of marker usage or histological patterns may pool data from heterogeneous populations (e.g., healthy individuals, patients with lipedema, cancer, or metabolic disease) without appropriate stratification. For example, the expression of markers such as CD34 or the degree of fibrosis might differ substantially in pathological tissue, yet this variability is not clearly addressed in the figures or narrative.
Additionally, differences between tissue obtained by surgical excision versus aspiration — which affect ECM composition and cell viability — are acknowledged but not sufficiently integrated into the comparative analysis.
To enhance the neutrality and generalizability of the review, the authors should more directly acknowledge and, where possible, analyze the impact of these confounding variables. A table summarizing population and sample source characteristics per study (when available) would significantly strengthen transparency and interpretive value.
7. Animal Studies and Sample Size Considerations
The authors are commended for clearly delimiting their review to human adipose tissue samples and excluding animal studies. However, given the widespread use of animal models in adipose tissue research—particularly in rodents—it would be helpful to briefly acknowledge this in the discussion and explain the rationale for their exclusion (e.g., interspecies differences in adipocyte size, immune microenvironment, and ECM composition).
Furthermore, while human sample size was not an exclusion criterion, the variability in study size was not systematically addressed. This raises the possibility that trends presented in the review may be disproportionately influenced by smaller or underpowered studies. Stratifying or at least acknowledging this variation would enhance the transparency of the synthesis and help readers better interpret the strength of evidence behind particular staining choices.
8. Impact of Sample Collection Method on Histological Technique
Although the review acknowledges different preparation types (e.g., microfat, lipoaspirates, surgical excision), the method of sample collection was not used as a formal inclusion or exclusion criterion, nor was it systematically analyzed as a factor influencing staining choice or interpretability.
This is a missed opportunity, as the collection method directly affects the histological features and the suitability of different stains. For instance, lipoaspirated tissue tends to show more adipocyte disruption and oil cysts, potentially requiring perilipin or Oil Red O for assessing adipocyte integrity, whereas excised tissue preserves architecture better and may be more suitable for fibrosis scoring with Masson’s Trichrome. These distinctions should be more clearly discussed, and the authors could enhance the practical utility of the manuscript by summarizing stain/tissue compatibility in a comparative table or figure.
9. Missing Support for Claims on Methodological Reporting and Tissue Origin
The authors argue that poor reporting of sample preparation and processing methods impairs reproducibility and comparability across studies. However, this important limitation is not adequately supported by data in the Results section. The manuscript does not report how many studies failed to describe key aspects such as fixation method, embedding, section thickness, or preservation conditions. If poor reporting is to be highlighted as a major issue, this should be substantiated by quantitative analysis in the Results section.
Additionally, in the Discussion, the authors propose that adipose tissue origin (e.g., surgical excision vs. aspiration) affects histological outcomes and staining reliability. While this is likely true and supported by previous literature, the review provides no indication that this variable was collected or analyzed as part of the data extraction process. The topic is absent from the Materials and Methods section, and it is unclear how many of the included studies reported on sample origin or how this may have influenced technique choice.
To ensure the conclusions are evidence-based, the authors should either: (a) explicitly state that tissue origin was extracted and present this analysis in the Results, or (b) clarify that these are interpretative hypotheses, not empirical findings from the reviewed literature.
II - Specific Comments
- Some arrangements must be made in the information pertaining to M&M and that of Results – some conflicting positioning exists in the current MS
- Lines 41–47: The abstract conclusion should better reflect the nuanced context in which complementary stains are useful and clarify their experimental vs. clinical roles.
- Lines 236–244: Consider qualifying that Masson’s Trichrome is primarily used in research settings for fibrosis quantification.
- Lines 282–292: While IHC markers for inflammation are valuable for specific immune phenotyping, H&E alone is generally sufficient for identifying inflammatory infiltrates in clinical practice.
- Figures and Tables: While informative, some graphs lack axis labels and legends. Consider improving the visual clarity and providing more detailed captions.
- The Prisma Flow Diagram must be corrected to include the excluded duplicated records.
- The authors do not present the list of retrieved and analyzed papers, which is unacceptable. They declare to have revised 166 papers, but they were not identified, and the reference list provided only includes 105 sources. No supplementary material was found attached to this submission.
- Data Availability Statement: The authors state "not applicable," but a PRISMA-compliant extraction table would strengthen reproducibility and transparency.
Final Recommendation: Major Revision
While the topic of this scoping review is both timely and important, and the overall structure follows accepted reporting guidelines, significant methodological issues limit the reliability and transparency of the findings. Specifically, there is a lack of alignment between the reported methods and the conclusions drawn. Key variables central to the discussion (e.g., tissue source, processing steps) are not systematically extracted or analyzed. The review also includes interpretative commentary that appears speculative or unsupported by the data presented. In its current form, the manuscript does not meet the standards expected for publication in a high-impact journal without substantial revision. Therefore, I recommend major revisions, including clearer methodological descriptors, evidence-backed discussion, and improved data synthesis and reporting transparency.
Additional comments and suggestions can be found in the attached copy of the MS

Author Response
Answer to Reviewer 1
We would like to thank you for your detailed and insightful feedback. We greatly appreciate the thoughtful critique and constructive suggestions, which have helped us improve the clarity, methodological transparency, and practical relevance of our manuscript. Below, we respond to each point raised and describe the corresponding revisions. All changes have been implemented in the revised manuscript, with marked text in yellow.
- Reviewer comment:
Weaknesses such as incomplete reporting, lack of standard preparation protocols, and low use of advanced imaging are only descriptively addressed. A structured assessment of methodological quality (e.g., using a formal appraisal tool) would strengthen the analysis.
Response:
We appreciate the reviewer’s suggestion regarding the assessment of methodological quality. While scoping reviews do not formally require risk of bias assessment or quality appraisal (as per JBI guidance), we agree that greater methodological transparency improves the utility of the synthesis. We have therefore included a new comprehensive data extraction Table 2 that documents which staining methods were used in each study and what they evaluated using these staining. This table provides a structured overview of histological and immunohistochemical techniques across all 166 included studies and allows readers to trace methodological trends. Additionally, the Results section refers to key trends observed in staining practices, supported by this table.
- Reviewer comment:
The authors effectively identify the absence of standardized staining protocols and inconsistent reporting as critical gaps. They offer constructive suggestions, such as adopting semiquantitative scoring systems and encouraging collaboration with pathologists. Proposals for increased use of digital tools and multiplex staining are appropriate and forward-looking.
Response:
We thank the reviewer for this positive and encouraging comment. In line with the suggestion to emphasize forward-looking developments, we have slightly expanded the Discussion section to acknowledge the potential of digital tools. These technologies may help address current limitations in reproducibility and allow for more comprehensive tissue analysis. We hope this addition will enhance the translational relevance of our review. Sine we already had a paragraph about the use of multiplex staining we highlighted it in red and added the term “multiplex staining” to acknowledge the scientifically correct name for what we described
- Reviewer comment:
While the authors provide a helpful quantitative overview of the most frequently used histological and immunohistochemical techniques, the rationale behind the selection of these techniques in the primary studies is not systematically explored. It remains unclear whether the prevalence of certain stains or markers (e.g., H&E, Masson’s Trichrome, CD31, CD34) reflects their technical suitability, accessibility, historical use, or methodological advantages.
Without this context, it is difficult to interpret whether frequently used techniques are also the most appropriate for the intended histological evaluation. To strengthen the interpretive value of the review, the authors should include a critical discussion on why these techniques might be preferred, supported by methodological or functional justifications where available. A comparative table linking techniques to their intended purpose, strengths, and rationale (when reported) would enhance the transparency and utility of the findings for researchers selecting appropriate tools for future studies.
Response:
We thank the reviewer for this important and detailed comment. In the revised manuscript, we now explicitly refer to Table 2, which summarizes the histological and immunohistochemical techniques used in the included studies alongside the tissue features they were applied to assess. We believe this table already provides a meaningful link between staining methods and their practical purpose.
Additionally, we have expanded the Discussion section to briefly address common reasons why specific stains and markers may be chosen, including their suitability for assessing adipocyte morphology, fibrosis, vascularization, or immune infiltration. To further clarify this, we have added the following sentence to the Discussion:
“This overview of stain-to-feature associations helps readers understand the methods used and why they are selected for specific histological evaluations.”
We hope these additions improve the interpretive depth and practical value of the review.
- Reviewer comment:
While the manuscript provides a comprehensive overview of staining techniques used in adipose tissue research, it would benefit from a clearer distinction between stains required for diagnostic histopathology and those employed in experimental or research contexts. Hematoxylin-eosin (H&E) remains the gold standard for evaluating adipose tissue morphology, inflammatory infiltrates, and structural lesions in clinical diagnostic practice. Experienced pathologists can readily identify adipocytes, fibrous tissue, necrosis, and inflammatory cells using H&E alone. Thus, complementary stains like Masson’s Trichrome, Oil Red O, or immunohistochemical markers such as CD68 or F4/80 are not generally required unless a specific diagnostic challenge arises.
In contrast, in regenerative medicine and adipose tissue engineering, additional staining techniques serve an investigative rather than diagnostic purpose. Masson’s Trichrome is used to quantify fibrosis; immunomarkers allow phenotyping of SVF or immune cells; and lipid stains help visualize oil cysts or damaged adipocytes. These methods are essential for quantitative and mechanistic insights and should be clearly contextualized as such. We recommend that the authors clarify these differences to avoid conflating clinical and experimental goals.
Response:
We thank the reviewer for this important and well-articulated comment. In the revised version of the manuscript, we have clarified the distinction between diagnostic and research-related staining techniques at multiple points:
- In the Abstract, we now emphasize that the focus of this review is on techniques used in experimental and translational research, not routine clinical diagnostics.
- In the Introduction, we specify that while H&E is a clinical gold standard, the reviewed staining approaches extend into research domains where additional techniques are applied to assess functional and structural parameters.
- In the Discussion, we now explicitly differentiate between diagnostic practice—where H&E is often sufficient—and the needs of regenerative medicine and tissue engineering research, which often require specialized stains and immunomarkers.
We hope these revisions provide the necessary clarity and help readers understand the scope and purpose of the staining techniques addressed in our review.
- Reviewer comment:
Another aspect deserving further elaboration is the timing of histological/IHC assessment and its impact on the types of data that can be retrieved.
Pre-inoculation analysis is valuable for evaluating graft quality and cell composition but requires sacrificing part of the material—making it suitable only for preclinical studies or tissue processing validation. This form of analysis informs the baseline characteristics of the graft but must be interpreted with care, as processing can introduce variability.
Post-inoculation assessment is more common in animal studies or when clinical tissue is re-biopsied. It allows analysis of graft integration, inflammatory responses, fibrosis, and cell survival—critical for interpreting treatment outcomes. However, the sampling time and method can affect tissue representativeness and comparability.
Moreover, destructive histological processing inherently compromises cell viability. To mitigate this, researchers often split the graft into aliquots (for histology vs. implantation) or rely on surrogate measures such as flow cytometry, metabolic assays, or non-destructive imaging (e.g., confocal microscopy).
We suggest the authors expand their discussion to acknowledge these limitations, as they directly shape the kind of histological evidence available in the literature and help readers understand why certain approaches are over- or under-represented.
Response:
We thank the reviewer for highlighting this important methodological consideration. In the revised version of the manuscript, we have expanded the Recommendations and operative considerations section to include a dedicated paragraph that outlines how the timing of histological analysis—before or after graft inoculation—affects both the feasibility and interpretive value of the data. We emphasize that pre-inoculation assessments are useful for analyzing baseline tissue characteristics but may not reflect how tissue behaves in vivo, while post-inoculation analysis allows evaluation of integration, inflammation, and remodeling processes, albeit with variability due to sampling conditions.
In addition, we briefly mention that certain studies address these limitations by using split samples or alternative non-destructive methods such as imaging or flow cytometry. We hope this addition addresses the reviewer’s concern and improves the methodological transparency of our discussion.
- Reviewer comment:
While the authors have limited their review to human studies and attempted to categorize adipose tissue by preparation type (e.g., lipoaspirate, microfat), the influence of external factors such as donor age, anatomical site, pathological status, or tissue processing method is not fully accounted for. These factors can significantly impact histological architecture, marker expression, and immune cell infiltration — yet appear to be unevenly reported across the included studies.
Moreover, some summaries of marker usage or histological patterns may pool data from heterogeneous populations (e.g., healthy individuals, patients with lipedema, cancer, or metabolic disease) without appropriate stratification. For example, the expression of markers such as CD34 or the degree of fibrosis might differ substantially in pathological tissue, yet this variability is not clearly addressed in the figures or narrative.
Additionally, differences between tissue obtained by surgical excision versus aspiration — which affect ECM composition and cell viability — are acknowledged but not sufficiently integrated into the comparative analysis.
To enhance the neutrality and generalizability of the review, the authors should more directly acknowledge and, where possible, analyze the impact of these confounding variables. A table summarizing population and sample source characteristics per study (when available) would significantly strengthen transparency and interpretive value.
Response:
We thank the reviewer for this important observation. As noted, we limited our review to human studies and categorized adipose tissue by preparation type (e.g., centrifugation, sedimentation, excised tissue), which we believe helps reduce methodological heterogeneity. However, we agree that additional factors—such as donor age, anatomical location, pathological background, and processing technique—can influence histological outcomes.
To address this, we have expanded the Discussion to include a specific paragraph acknowledging the limited and uneven reporting of such variables in the included studies. While detailed stratification was not feasible due to inconsistent data availability, we now explicitly describe this limitation and caution against overinterpretation of pooled results across heterogeneous populations.
We believe this addition enhances the interpretive transparency of the review.
- Reviewer comment:
The authors are commended for clearly delimiting their review to human adipose tissue samples and excluding animal studies. However, given the widespread use of animal models in adipose tissue research—particularly in rodents—it would be helpful to briefly acknowledge this in the discussion and explain the rationale for their exclusion (e.g., interspecies differences in adipocyte size, immune microenvironment, and ECM composition).
Furthermore, while human sample size was not an exclusion criterion, the variability in study size was not systematically addressed. This raises the possibility that trends presented in the review may be disproportionately influenced by smaller or underpowered studies. Stratifying or at least acknowledging this variation would enhance the transparency of the synthesis and help readers better interpret the strength of evidence behind particular staining choices.
Response:
We thank the reviewer for this helpful suggestion. In the revised Discussion section, we highlighted the part were we explained out decision to exclude animal studies from our analysis in red. We explained that we deliberately excluded non-human studies to maintain relevance to clinical and translational applications. We note that species-specific differences in adipocyte morphology, extracellular matrix composition, and harvesting technics limit the comparability of animal and human tissue.
Additionally, we have added a sentence highlighting that sample sizes varied substantially among the included studies, and that this heterogeneity may affect the interpretability of technique frequency and staining outcomes. As many studies did not report clear sample size thresholds or power calculations, we treated this as a limitation of the underlying literature and have cautioned readers accordingly.
- Reviewer comment:
Although the review acknowledges different preparation types (e.g., microfat, lipoaspirates, surgical excision), the method of sample collection was not used as a formal inclusion or exclusion criterion, nor was it systematically analyzed as a factor influencing staining choice or interpretability.
This is a missed opportunity, as the collection method directly affects the histological features and the suitability of different stains. For instance, lipoaspirated tissue tends to show more adipocyte disruption and oil cysts, potentially requiring perilipin or Oil Red O for assessing adipocyte integrity, whereas excised tissue preserves architecture better and may be more suitable for fibrosis scoring with Masson’s Trichrome. These distinctions should be more clearly discussed, and the authors could enhance the practical utility of the manuscript by summarizing stain/tissue compatibility in a comparative table or figure.
Response:
We thank the reviewer for highlighting this important methodological aspect. In the revised manuscript, we added a table summarizing the recommended stains, their architecture, target, and specific characteristics/limitations in the "Recommendations and Operative Considerations" section. For example, lipoaspirates often present with disrupted adipocytes and oil cysts, making them more compatible with lipid stains such as Oil Red O or perilipin. In contrast, excised tissue retains architectural features and may be better suited for assessing fibrosis using Masson’s Trichrome.
We hope this improves the practical relevance of the manuscript for researchers selecting staining protocols based on tissue type.
- Reviewer comment:
The authors argue that poor reporting of sample preparation and processing methods impairs reproducibility and comparability across studies. However, this important limitation is not adequately supported by data in the Results section. The manuscript does not report how many studies failed to describe key aspects such as fixation method, embedding, section thickness, or preservation conditions. If poor reporting is to be highlighted as a major issue, this should be substantiated by quantitative analysis in the Results section.
Additionally, in the Discussion, the authors propose that adipose tissue origin (e.g., surgical excision vs. aspiration) affects histological outcomes and staining reliability. While this is likely true and supported by previous literature, the review provides no indication that this variable was collected or analyzed as part of the data extraction process. The topic is absent from the Materials and Methods section, and it is unclear how many of the included studies reported on sample origin or how this may have influenced technique choice.
To ensure the conclusions are evidence-based, the authors should either: (a) explicitly state that tissue origin was extracted and present this analysis in the Results, or (b) clarify that these are interpretative hypotheses, not empirical findings from the reviewed literature.
Response:
We thank the reviewer for raising this important point and appreciate the opportunity to clarify our approach.
As noted, we have now added Table 2, which provides a detailed summary of the tissue preparation type (e.g., lipoaspirate vs. excised fat) used in each included study. We also report in the Results section that 25 of the 166 studies did not provide any information regarding their preparation technique, which further underscores the challenges in comparing histological results across the literature.
However, we did not extract data on fixation method, embedding protocol, section thickness, or preservation conditions, as these aspects were not the primary focus of our review. These procedures are largely standardized in histopathology, and our goal was to concentrate specifically on the choice of histological stains and the types of biological or structural information extracted from them.
Regarding the influence of tissue origin and preparation on stain selection and interpretability, we refer the reviewer to the newly added Table 3, which synthesizes our findings on stain suitability in relation to tissue type and target features. This table was created in response to another reviewer comment and addresses the present concern by summarizing how methodological differences can impact staining outcomes.
We trust that this clarification, together with the new tables and textual revisions, adequately addresses the reviewer’s comment and strengthens the transparency and empirical grounding of our conclusions.
II - Specific Comments
Lines 41–47: The abstract conclusion should better reflect the nuanced context in which complementary stains are useful and clarify their experimental vs. clinical roles.
Response: We specified our conclusion in the abstract according to this recommendation
Lines 236–244: Consider qualifying that Masson’s Trichrome is primarily used in research settings for fibrosis quantification.
Response: We specified this part according to this recommendation
Lines 282–292: While IHC markers for inflammation are valuable for specific immune phenotyping, H&E alone is generally sufficient for identifying inflammatory infiltrates in clinical practice.
Response: An explanation was added to clarify that those markers in IHC are only important in scientific research and not in clinical routine.
-The Prisma Flow Diagram must be corrected to include the excluded duplicated records.
Response: The PRIMSA Chart was corrected according to the PRISMA Guidelines
Figures and Tables: While informative, some graphs lack axis labels and legends. Consider improving the visual clarity and providing more detailed captions.
Response: The Figures where corrected due to the considerations of the Reviewer.
Reviewer: The authors do not present the list of retrieved and analyzed papers, which is unacceptable. They declare to have revised 166 papers, but they were not identified, and the reference list provided only includes 105 sources. No supplementary material was found attached to this submission.
Response: All those 166 Papers can now be found in Table 2 and in the Bibliography.
Data Availability Statement: The authors state "not applicable," but a PRISMA-compliant extraction table would strengthen reproducibility and transparency.
Response: The Data Availability Statement is now corrected and PRISMA compliant
Line in the Original Manuscript |
Comment of Reviewer 1 |
Response |
||
37 |
"distinguish ?" |
Evaluate was changed to distinguish |
||
55 |
|
Reference 1 and 2 were added |
||
65 |
"please add the appropriated references after each type of markers used in the studies" |
References 8-10 were rearranged |
||
120 |
“but no manual searchers were performed in other databases. Why not, if in theory they would be equivalent?” |
We added the sentence In our opinion those 3 databases fit the research question the best |
||
130 |
"why? if the aims are the histology technique, they are similar in human and veterinary medicine domains.
However, if the focus was the identification of specific features, then the exclusion could be necessary” |
The answer can be found in the original manuscript in the lines 338-352 and is highlighted in red in the revised version. And we added tissue to retain the focus to identification of specific features |
||
132-134 |
"why is this important??" |
We added each dataset is only included once. To clarify why this is important as the sentence before was not clear in this point |
||
150 |
“Table number?” |
We added Tab. 1 Inclusion and Exclusion Criteria of this Scoping Review as a Titel for the Table 1
|
||
152 |
"sections 3.1, 3.2 and 3.3 should be moved to Material and methods. The results section should include only the information retrieved to answer your reseacrh question about methos for adipose tissue histological evaluation" |
The sevtions 3.1, 3.2 and 3.3 were moved to the Material and Methods |
||
153 |
"you mentioned a search through the documents reference list. how does it enter in here?" |
We added the Results from the Manual reference search into the PRISMA chart |
||
160 |
“a list of papers retrieved is recommended, to be presented as a supplementary file.” |
The references were added to Table 2 instead of a new Table in the supplementary Materials |
||
161 |
"Why this subsection here? didn’t you exclude all animals samples (even though not explaining adequately the reasons)?" |
We added We excluded all studies that featured only animals, but we included those that focused on the transplantation of human adipose tissue into animals. To make matters clearer in this case. |
||
171 |
“was the reason for it mentioned? Were you able to justify the rationale behind the selection?” |
In our section Discussion we justify the rationale behind the reason as of why to use certain histological stainings and IHC and in Recommendations and operative considerations we added Table 3 to summarize these findings |
||
180 |
“similar comment applies here: why were these markers used, and not others? Propabily it respect the purposed for the adipose tissue histological analysis. but as it determined also the selection of biomarkers used, the reasons should be identified” |
|||
185 |
"how did the preparation methods affected the selected histological methods?" |
We added While this underscores the primary rationale behind our emphasis on the selection of the preparation protocol, it should be noted that no correlation exists between the choice of staining and the preparation method. |
||
195 |
"were the qualitative markers used at specific moments in the adipose tissue evaluation (e.g., before implantation/introduction; after negative outcomes?; ...)" |
We added Table 3 in which we recommended when to use which Marker. Also we think that the exact answer to this very good question is in Line 382-384 of the original manuscript |
||
203 |
"no need to add a new section, as this topic respects qualitative studies" |
We combined the subsection with 3.7 to reduce the fractioning of the text |
||
208 |
|
|||
217 |
“No need to provide a subsection here. the text herin can be moved to before the headings “histological techniques”” |
The subsection headline was deleated and the text was shifted to the headline “Histological Techniques” |
||
226 |
“n=??” |
We added the N under Figure 3 : Single-marker immunohistochemistry was used in 33 studies and multiparametric techniques 57 times while in 76 studies no immunohistochemistry was assessed.
|
||
227 |
“which variations? Should be detailed” |
We added detailed information about the variation of preparation methods in Table 2 |
||
232 |
“Is this results? Or should it be discussion??” |
We shifted the subsections 3.13 and 3.14 to the discussion |
||
247 |
“What are the strengths and weaknesses between the mentioned stainings and the objectives for their use?” |
the strengths and weaknesses of the staings were now listed in Table 2 and described in detail in the section above this Table. |
||
260 |
“two ??” |
The misspelling of the word two was corrected |
||
282 |
"is side effect the best term?" |
The Term side effect was changes to complications |
||
287 |
"only macrophages? other inflammatory cells are not important?" |
Macrophages are in most cases the only inflammatory cell that is observed. However we made the text more clear to ensure that in theory other cells could be evaluated too. |
||
290 |
"then I can not understand the exclusion of animal/veterinarian papers from this review" |
After rereading this passage, we found it confusing to. We did not wanted to emphasize that we mainly used studies with Animal fat, since we excluded them and gave a firm argumentation about why we did that. We instead wanted to acknowledge that in most studies Animals were set up for explantation with Xenografts from Human origin and those studies are of course not excluded. We made sure the passage is now correct. |
||
298 |
|
We added citation 200 and 201 to support our synthesis |
||
299 |
"include in this section the text where the different staining methods are discussed" |
We shifted the subsections 3.13 and 3.14 to the discussion |
||
329-334 |
|
We did completely restructure this paragraph because as the Reviewer emphasized it had a lot of potential misunderstandings and even mistakes. We pointed out what we meant with preparation and processing methods and clearly defined them. We also cut the last part as we were not able to give proper explanation and thus were highly speculative in our assumption about interstudy comparability and reproducibility. |
||
|
|
|||
334 |
"how did you reach to this conclusion. no data is reported. This sentence is highly pseculative!" |
|||
344-352 |
|
Unfortunately the comment pinned on this part of the Review is empty. As the topic of Animal studies was addressed before we can only assume that the author had an further idea about this subject. After thorough proofreading of this paragraph by us we must confess that we do not know were the mistake the Reviewer considered might be. Therefore we did not change the paragraph even thou we highlighted it red because of another comment mentioned before. |
||
356 |
|
In our opinion this provides good proof that the Reviewer deeply understood the point we were trying to make in this scoping review. Unfortunately we were not able to articulate it precisely enough. For clarification we added the sentence Furthermore, the technician and the laboratory equipment could influence the methodologies and results of histology. Therefore, in the future, a certification program for providers of histological techniques should be developed. |
||
364-365 |
|
As the Reviewer pointed out, we really did not provide a comprehensive guide on the complete planning and execution of histological analysis for scientific research projects. So, we decided to cut the whole sentence to not be misleading. We even added a section where we defined this limitation of our synthesis to be as transparent as possible. It can be found right above the paragraph that starts with “The exclusion of studies utilizing whole…” |
||
374-375 |
"have you provide this information on results?" |
We did in line 188-189 in our original manuscript |
||
398 |
"this information does not cope with the discussion section. may be you could introduce a new section titled as recommendations, operative considerations, or similar?" |
We think this can be strengthening for better reading flow and therefore we added the new subsection Recommendations and operative considerations |

Reviewer 2 Report
Comments and Suggestions for Authors
I have carefully read the manuscript cells-3644574.
The review article is very focused, with a clear introduction that leads well into the study's objective.
The Materials and Methods section is concise and clear, particularly regarding the inclusion and exclusion criteria.
However, I would like to ask: What were the reasons for not using other common sources such as Scopus or Web of Science?
Additional questions for this section: How was the frequency of use of the different staining methods defined and measured?
The presentation of the results is clear, although in one of the figures, “Inmunohistology” should be corrected to “Immunohistochemistry.”
It would also be important to clarify the abbreviations used for the antibodies. Even if they are somewhat intuitive, defining them is necessary—especially if some are defined and others are not. Consistency is essential.
The discussion is appropriate and focuses on the results. Some questions to consider:
What are the practical implications of these findings for regenerative medicine?
How can these results guide the selection of staining methods in future studies?
The conclusion is appropriate.
Keywords should differ from the words used in the title to increase the chances of the article being found.
The findings have significant value, particularly for the study of human adipose tissue
Author Response
Answer to Reviewer 2
We thank you for your constructive and supportive feedback. We appreciate the clear reading of our manuscript and the valuable suggestions, all of which helped to further improve the clarity, methodological transparency, and practical relevance of our work. Below we provide point-by-point responses to each comment, along with corresponding revisions to the manuscript highlighted in a yellow hue.
- Reviewer comment:
“What were the reasons for not using other common sources such as Scopus or Web of Science?”
Response:
Thank you for this observation. In the revised Materials and Methods section, we now clarify that PubMed was selected due to its specificity for biomedical literature and alignment with JBI scoping review methodology. We acknowledge that excluding Scopus and Web of Science may have limited the comprehensiveness of the search. But with 1122 Articles being reviewed we are optimistic that we made a broad and thoughtful scope above the literature.
- Reviewer comment:
“How was the frequency of use of the different staining methods defined and measured?”
Response:
Thank you for this helpful request for clarification. We have specified in the Methods section that frequency was defined as the number of included studies reporting a given staining technique, based on explicit mention in the original article. No weighting was applied for sample size or design complexity. In our discussion, we added a paragraph highlighting the limitation of our study in this aspect.
- Reviewer comment:
“…‘Inmunohistology’ should be corrected to ‘Immunohistochemistry.’”
Response:
We thank the reviewer for spotting this error. The term “Immunohistology” in the figure has been corrected to “Immunohistochemistry.”
4.Reviewer comment:
“It would also be important to clarify the abbreviations used for the antibodies. Even if they are somewhat intuitive, defining them is necessary.”
Response:
Thank you. We have reviewed all abbreviations for immunohistochemical markers and ensured that they are consistently defined upon first mention.
- Reviewer comment:
“What are the practical implications of these findings for regenerative medicine?”
Response:
We appreciate this suggestion. In the revised Version we made sure to point out that our findings are important for regenerative medicine because of its underlaying importance for scientific research of the field. We added new phrases and paragraphs in the Discussion emphasizing the relevance of Histology for the scientific research and the connection towards more specialized questions depending different pathologies as lipedema or metabolic disorders.
6.Reviewer comment:
“How can these results guide the selection of staining methods in future studies?”
Response:
Thank you. We have expanded the Discussion with the subsection Recommendations and operative considerations to suggest how the reported association between staining methods and target tissue features may help researchers align their technique choices with specific investigative aims. Table 3 supports this by linking stains to their analytical applications.
7.Reviewer comment:
“Keywords should differ from the words used in the title to increase the chances of the article being found.”
Response:
Thank you for this editorial suggestion. We have revised the keywords to increase the chance of the article being found and include more search-relevant terms such as: “regenerative medicine”, “lipoaspirate”, and “staining protocol.”
We thank the reviewer once again for the thoughtful input and hope that the revised version of the manuscript addresses all concerns satisfactorily.

Reviewer 3 Report
Comments and Suggestions for Authors
This manuscript, a review focusing on immunohistochemistry in adipose tissue, is found to be notably deficient in the presentation of scientific significance supported by biological data, as well as in providing adequate experimental methodological details. As a result, it offers limited value to the adipose tissue biology community, lacking crucial insights into both biological implications and practical methodologies. Based on these limitations, the manuscript is deemed not suitable for publication.
Author Response
Answer to Reviewer 3
We thank you for taking the time to review our manuscript. We respectfully disagree with the assessment that the manuscript lacks scientific significance and methodological clarity. This manuscript is designed and structured as a scoping review, following PRISMA-ScR and JBI guidelines. Its objective is not to present novel experimental data, but rather to provide a thoroughly scope of histological and immunohistochemical staining techniques applied in human adipose tissue research.
While we do not include primary biological experiments, our work offers a comprehensive synthesis of 166 studies, highlighting commonly used staining protocols, immunomarkers, and their corresponding tissue features. We also discuss methodological variability, reporting gaps, and propose practical recommendations aimed at improving reproducibility and standardization in adipose tissue analysis.
Furthermore, in the revised version, we have:
- Clearly clarified the scope and aim of the review in the Introduction. “The goal of this scoping review is not to generate new biological hypotheses, but rather to consolidate existing methodological practices in adipose tissue histology and immunohistochemistry thereby identifying patterns, gaps, and areas for improvement”
- Strengthened the Conclusion section to better highlight the practical and translational relevance of our findings to regenerative medicine and tissue engineering. “The findings of this review are a valuable reference for researchers planning a histological evaluation of adipose tissue in the field of regenerative medicine and tissue engineering. They also support the idea of improving methodological consistency in clinical and translational contexts.”
- We hope that these clarifications and additions demonstrate the manuscript’s relevance to the adipose tissue biology community and address the reviewer’s concerns.

Round 2
Reviewer 1 Report
Comments and Suggestions for Authors
The authors addressed my previous concerns in full in the resubmitted manuscript. Two points were only partially addressed, and additional minor corrections or clarifications remain.
The first respects the topic of structured assessment of methodological quality. Although inconsistencies and reporting quality are now better quantified, a formal appraisal tool (e.g., adapted JBI checklist, scoring rubric) has still not been applied. However, the scoping nature of the review may justify this omission. But across the MS, we learn from the text that some pitfalls seem to exist in the retrieved and analyzed articles that limit the author's analysis (e.g., the clarification of the protocols). We still don´t know which studies were at fault for the critical information in the new text. It could be helpful to have a table highlighting the characterization of defaulted studies, according to you, as supplementary material. You mention it in the text, but do not provide details, which is crucial information for the final staining output.
Secondly, the topic of population heterogeneity was also only partially addressed. The authors acknowledged variability due to age, pathology, and collection site. However, these variables were not extracted due to inconsistent reporting across primary studies. This is transparently noted as a limitation but reinforces the need for a supplementary table identifying critical information in the analyzed papers.
In addition, minor changes are needed (font type or size; additional clarification on the title; text alignment, correction of captions, among others). They are signaled in the received file, for details.

Author Response
Answer to Reviewer 1
We would like to sincerely thank you once again for your exceptional level of detail, expertise, and dedication throughout the review process. Your comments have not only significantly improved the clarity, methodological transparency, and academic rigor of our manuscript but have also challenged us to think more critically about the limitations and interpretive scope of our work.
The depth of your feedback, ranging from conceptual nuances to formatting details, has helped us refine our review on multiple levels. Your insistence on methodological transparency and interpretive clarity pushed us to extract the best possible version from a previously rough and unfocused draft. We are especially grateful for the constructive tone of your critique, which made it clear that your goal was to enhance the scientific value and practical utility of this paper.
We truly appreciate the time, effort, and scholarly care you invested. Your input was instrumental in transforming our manuscript into a far more coherent, precise, and useful contribution to the field. All changes done are marked in green to not confuse them with the corrections from the first version of the Manuscript.
Reviewer comment:
The first respects the topic of structured assessment of methodological quality. Although inconsistencies and reporting quality are now better quantified, a formal appraisal tool (e.g., adapted JBI checklist, scoring rubric) has still not been applied. However, the scoping nature of the review may justify this omission. But across the MS, we learn from the text that some pitfalls seem to exist in the retrieved and analyzed articles that limit the author's analysis (e.g., the clarification of the protocols). We still don´t know which studies were at fault for the critical information in the new text. It could be helpful to have a table highlighting the characterization of defaulted studies, according to you, as supplementary material. You mention it in the text, but do not provide details, which is crucial information for the final staining output.
Secondly, the topic of population heterogeneity was also only partially addressed. The authors acknowledged variability due to age, pathology, and collection site. However, these variables were not extracted due to inconsistent reporting across primary studies. This is transparently noted as a limitation but reinforces the need for a supplementary table identifying critical information in the analyzed papers.
Response:
We thank you for the thoughtful and constructive suggestion. As noted, scoping reviews, according to JBI and PRISMA-ScR guidance, do not mandate a formal appraisal of methodological quality. We fully agree, however, that insufficient reporting and methodological variability in the included literature pose limitations, particularly regarding standardization of staining protocols and interpretation of results. Nonetheless, a decision was taken to undertake a systematic analysis of the factors in question to address the fact that these differences in the histological outcome are very well understood, as was highlighted in the synthesis.
There is also a comment on Page 17 we addressed in the end of our answer regarding the same issue and furthermore clarifies which studies are not reporting their preparation protocol for fat tissue and where they can be found.
Whilst no new table was designed to address these informations, it was clarified in which sections and tables the reader could find such information.
To the second point we appreciate your continued attention to the role of population heterogeneity in shaping histological findings. As mentioned, we deliberately included only human studies and classified tissue samples by preparation method (e.g., centrifuged, mechanically processed, excised tissue), which we consider a meaningful way to reduce heterogeneity.
Moreover, we fully agree that biological variability related to donor age, disease status, and anatomical site can in theory influence tissue characteristics. In our reviewed articels, however, such variables were frequently underreported or ambiguously stated in the original articles, preventing reliable extraction or synthesis. For this reason, and to avoid presenting incomplete or misleading stratifications, we chose not to include a supplementary table summarizing these factors.
Instead, we have expanded the Discussion to more explicitly reflect this limitation and its implications for interpretation. We hope this approach preserves transparency while maintaining methodological integrity and avoiding overinterpretation of inadequately reported data.
Reviewer comment:
In addition, minor changes are needed (font type or size; additional clarification on the title; text alignment, correction of captions, among others). They are signaled in the received file, for details.
Response:
In the following we respond to every comment in the attached version of the manuscript that was given to us from you.
Page in the Original Manuscript |
Comment of Reviewer 1 |
Response |
1 |
“If translactional research is one of the goals for this review, it seems incongruous to eliminate articles from veterinary the medicine field“ |
In the last corrected Version of our Manuscript we explained that human fat which was transplanted into rodents were not eliminated from the research. Only studies that focused on fat that originates from rhodents was eliminated because of reasons we explained on Page 17 in the paragraph starting with :” The exclusion of studies utilizing whole …”. Because of this we think it is adequate if we say we try to support translational researchers. |
1 |
“Under the assumption that most research lab doing tranlational medicine work in rodents” |
|
1 |
“were these studies excluded? |
The 25 studies we found to not provide further information about the preparation methods of the Lipoaspirate were not excluded because they presented the staining techniques and histological evaluation to our satisfaction and therefore meet the criteria we saw as important for the evaluation process. We just want to highlight the fact that this lack of information makes studies difficult to compare even in case of a systematic review based on this very question of the influence of preparation protocols on the histological outcome. |
2 |
“usefull” |
We corrected the term “used” to the more fitting term “useful” |
3 |
“unless it respects variations in the staining protocol or timimgs” |
After rereading this section we also think that the paragraph was misleading in a certain way. So we decided to add “… of the fat tissue.” to clarify which type of “preparation method we meant” |
4 |
“include in here also the studies with animals transplanted with human fat” |
This is a very good point and should definitely belong into the inclusion criteria. So we added this aspect to the manuscript. |
5 |
“if you did not performed this type of search, it does not make sense to presente it here” |
We agree with your comment and therefore cut this part. |
5 |
“PRISMA 2020 flow diagram for the [give your target here] according to [27]” |
We agree that this sentence is better written than our original explanation, so we changed the sentence in accordance with the suggestion made. |
6 |
“instead of using the text in columns centered, use it in a left - alignment” |
As suggested from you we changed the alignment to left instead of center |
10 |
“revise the main title for the table. |
We revised the main title for the table and set the font size to 9 in the caption. |
10 |
“to separate the abbreviations use semicolon ( ; ) instead” |
We are now using semicolon to separate the abbreviations. |
10 |
“.” |
We added the missing “.”. |
10 |
“(i.e.,…)” |
We added the brackets and the i.e. to clarify our explanation as an example. |
10 |
“of animals used for human transplanted fat tissue was...” |
We added the partial sentence to our manuscript for better clarification |
11 |
“font size!” |
We changed the font size of this caption to 9 |
12 |
“font” |
We changed the font size of this caption to 9 |
14 |
“This study showed that” |
We think changing this passage towards the comment makes it more clear so we changed the sentence in accordance your comment. |
17 |
“add ref” |
We added reference 147 to the manuscript. |
17 |
“add the number of the source” |
We added the number of the citation. |
17 |
“we don´t know which studies were at fault for critical information. |
We absolutely agree with your comment. The information that is being asked is well hidden in Table 2. Within this table all of the 25 studies in question were marked with “-” for “No comment” we now clarified what that means in the caption of Table 2 so the studies in question can be found by readers. |
17 |
“refs” |
We added the reference according to the comment. |

Reviewer 3 Report
Comments and Suggestions for Authors
Extensive revisions have been made, significantly improving the quality of the paper.
Author Response
Answer to Reviewer 3
Reviewers comment:
Extensive revisions have been made, significantly improving the quality of the paper.
Response:
We would like to express our heartfelt thanks to you for your thoughtful and supportive feedback. We are truly grateful for your time and effort in reviewing our manuscript. Your concise yet impactful remarks underscored essential aspects of clarity, structure, and contextual framing, which helped us significantly refine the final version.
In particular, your suggestions contributed to a clearer definition of key terms and a more precise positioning of the review within the broader scientific landscape. This was instrumental in guiding us to make the scope and purpose of our work more immediately accessible to the reader, a key element for any successful scoping review.
We deeply value the peer review process as a cornerstone of scientific progress, and your contribution exemplifies how constructive input can elevate the quality and impact of a manuscript. Thank you again for your careful reading and for helping us bring this review to a much stronger and more coherent form.
